# GAA-PtrNet: Graph attention aggregation-based pointer network for one-shot DAG scheduling

## Abstract

Optimizing Directed Acyclic Graph (DAG) workflow makespan by scheduling techniques is a critical issue in the high performance computing area. Many studies in recent years combined Pointer Network (PtrNet) with reinforcement learning (RL) to schedule DAGs by generating DAG task priorities in a sequence-to-sequence manner. However, these PtrNet-based scheduling methods need to repeatedly compute the decoder's hidden state or context embeddings according to the recent local decisions, which leads to limited capability of exploiting the DAG global topological structure, high computation complexity and inability to achieve one-shot scheduling. To address these issues, we propose GAA-PtrNet, a novel PtrNet based on graph attention aggregation (GAA) for one-shot DAG workflow scheduling. In GAA-PtrNet, we compute the pair-wise graph attention scores among nodes in one-shot, then directly aggregate these scores to obtain the probability of decision sampling in sequence. Consequently, the explicit decoder or context embedding structure in PtrNet is omitted in our GAA-PtrNet, and the network takes only one shot forward propagation to infer a solution for a whole DAG scheduling problem, significantly reducing the computation complexity. Additionally, to train GAA-PtrNet, we design a training strategy based on policy gradient RL with dense reward signal and demonstration learning. To our knowledge, GAA-PtrNet is the first network model to achieve PtrNet-based one-shot DAG scheduling. GAA-PtrNet can better handle with DAG workflow structures, providing high quality DAG scheduling solutions. The experimental results show that the proposed method is superior in terms of objective and runs about 10 times faster when compared to previous PtrNet-based methods, and also performs better than other learning-based DAG scheduling methods.

## 1 Introduction

The Directed Acyclic Graph (DAG) scheduling problem arises in the high performance computing field (Hosseini Shirvani, 2024).It is a class of NP-hard Combinatorial Optimization Problems (COP), involving large and complex DAG workflow, and homogeneous or heterogeneous computation resources. In this domain, DAG is used to represent the parallel and sequential relationships among computation tasks. These tasks modeled as the nodes in the DAG, and the directed edges in the DAG represent the precedence constraints among the tasks. The goal is to achieve the best performance by determining an optimal node execution or priority order and allocating then to computation resources.

In recent years, machine learning, especially reinforcement learning (RL) technique, has already shown promise in DAG scheduling (Gu et al., 2025). A common network architecture for RL-based DAG scheduling consists of a Graph Neural Network (GNN) encoder to extract workflows' structural information, and a policy network to output scheduling decisions (Mao et al., 2019; Zhou et al., 2022; Song et al., 2023; Yu et al., 2023; Dong et al., 2023). These earlier DAG scheduling approaches follow Markov decision process and generate a solution step by step, requiring repeated extraction of global environment features and recalculation of schedules, which results in high computational overhead. The recent work of Jeon et al. (2023) addressed this issue by developing one-shot neural scheduler, which generates all the sub-decisions through a single forward pass of the

network. This one-shot GNN+RL method relies on generating a global logits list for all task nodes by the policy network, in which the Gumbel top-k trick (Kool et al., 2019) is introduced to perturb the global logits list. The logits list is treated as the task priority list to derive a task execution ordering via list-scheduling heuristics. However, the scheduling method which achieves ranking by generating a global list of logits inherently has large policy gradient variance. Additionally, such list-scheduling based on global logits list suffers from the fact that multiple distinct permutations of the logits list may correspond to the same valid schedule. This many-to-one mapping biases the probability of policy sampling, so that policy gradient estimation is biased in the learning process, which makes the scheduler training more prone to local optima. Qi et al. (2025) proposed a comparable antichain identification mechanism, from the perspective of reducing redundant pairwise comparisons among logits during ranking, partly addressed this issue. However, this method still depends on generating a global logits list to rank the task nodes.

Since DAG scheduling problem can be interpreted as a ordering problem over the problem components, Pointer Network (PtrNet) (Vinyals et al., 2015), as a sequence learning method, has shown its distinct advantages. Kintsakis et al. (2019) first introduces PtrNet into DAG workflow scheduling, which follows the foundational structure proposed by Bello et al. (2016). It encodes the task features with a long short-term memory (LSTM) encoder, and feeds the feature embeddings to an LSTM decoder. At each decision step, it computes the additive attention scores the current candidate actions according to the the decoder's hidden state, and the task with the highest attention score (i.e., the pointer) is selected for the following scheduling, thereby this method constructs the entire task priority sequence incrementally. Such similar PtrNet-based scheduling method is adopted by Dong et al. (2021); Zhao et al. (2022); Chen & Wang (2024); Li et al. (2022) for DAG scheduling. In contrast, Lee et al. (2020; 2021); Shi & Yu (2023); Wang et al. (2023), follows the improved PtrNet proposed by Deudon et al. (2018) and Kool et al. (2018). These works abandon calculating attention scores from LSTM decoder's hidden state. Instead, they take the context embedding of the current environment state and recent decisions to compute attention, partly addressing the limitations of traditional LSTM-based PtrNet scheduler for graphed structure problems. Both of these implementation do not rely on producing a global logits list for ranking, and therefore avoids the bias in solution-probability estimation that arises in ranking-based one-shot methods.

However, these PtrNet-based scheduling methods still suffer from limited capability of exploiting the global topological structure of DAGs, high time complexity and inability to achieve one-shot scheduling . As shown in Fig 1a, to construct a complete solution, PtrNet requires to repeatedly calculate the decoder hidden states or the context embeddings according to the observed environment state and recent decisions, and further obtain the attention scores of the candidate decisions according to these hidden states or embeddings. Consequently, their performance are often limited by their reliance on local information from recent decisions, which fail to capture long-range dependencies and the global topological structure inherent in the DAG. The repeated calculation of decoder's hidden states or context embeddings also leads to high computational complexity (Bello et al., 2016).

In this paper, we propose GAA-PtrNet, a novel PtrNet based on graph attention aggregation (GAA) for one-shot DAG scheduling, as shown in Fig 1b. Our GAA-PtrNet fundamentally addresses the above challenges from the perspective of omitting the ranking on logits lists and achieving a one-shot PtrNet-based scheduler. In general, the attention computation process obtains all priorities in one-shot, while task selection remains sequential, without the need for repeated computations of encoding and decoding of PtrNet. GAA-PtrNet consists of a trainable network to calculate pair-wise node attention scores in one shot, and a scheduler that aggregates attention scores to obtain sampling probabilities. In the network, a GNN encoder is employed to generate DAG task nodes' embeddings, then the pair-wise attention scores between all task nodes are calculated by graph attention mechanism. These two processes are achieved through a one-shot forward propagation of the network. In the scheduler, at each sequencing step the attention scores between the subgraph formed by the scheduled task nodes and each candidate task node are aggregated to calculate the probability to select candidate task nodes. Furthermore, to train GAA-PtrNet, we design a training strategy based on policy gradient RL. The contributions of this study are as follows:

1. We propose GAA-PtrNet, a novel PtrNet based on GAA for one-shot DAG scheduling. GAA-PtrNet innovatively aggregates the graph attention scores among task nodes after extracting the task node features and computing the graph attention in one-shot, which are directly used for calculating the policy sampling probabilities in stepwise, thereby realizes the one-shot PtrNet DAG scheduler.

GAA-PtrNet has strong capabilities to deal with workflow's topological structures with low time complexity. To the best of our knowledge, this is the first network model to achieve PtrNet-based one-shot DAG scheduling.

2. We design a training strategy based on policy gradient RL. We introduced dense reward signal and demonstration learning in training. Comprehensive experimental results show that the proposed method is superior in terms of makespan and runs about 10 times faster when compared to previous PtrNet-based methods, and also performs better than other learning-based DAG scheduling methods.

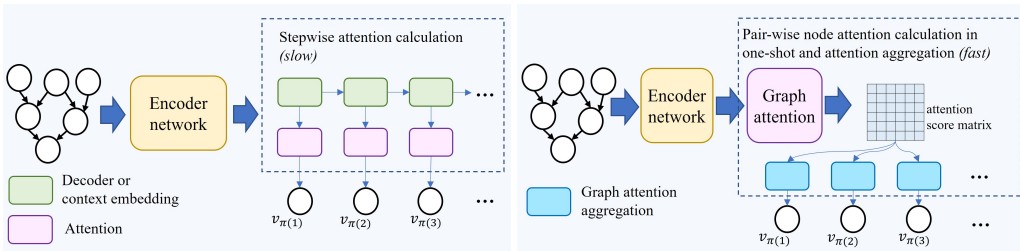

(a) Existing PtrNet.    (b) Our proposed GAA-PtrNet.

Figure 1: The illustrated comparison of existing PtrNet and our GAA-PtrNet in DAG scheduling. $v_{\pi(1)}, v_{\pi(2)}, ...$ represent the nodes scheduled at each steps.

## 2 PRELIMINARIES

### 2.1 DAG TASK MODEL

In a typical DAG scheduling problem $G = (V, E, X)$, the node set $V = \{v_1, v_2, ..., v_{|V|}\}$ represents the computation tasks. $X = \{x_1, x_2, ..., x_{|V|}\}$ is the attribution of each task node, primarily including the computational workload $c_i$, the output data size $b_i$, etc. The directed edges $E \subseteq V \times V$ denotes the precedence constraints among tasks. If $(v_i, v_j) \in E$, $v_j$ cannot start until $v_i$ is completed. For the convenience of calculation, a pseudo entrance task node will be created for the whole DAG scheduling problem, with all the nodes without predecessor to be its successors. Each task $v_i$ is associated with a processing time $d_i$. Some DAG systems require to consider the data transmission time $(z_i)$ between processors. A task node can be ready and executed only after all its predecessors' execution and transmission is finished.

### 2.2 DAG SCHEDULING

We consider work conserving scheduling: when existing ready tasks and an idle processor, the ready task with the highest priority is immediately selected and assigned to the processor. Determining the priority topological sorting $Solution(G) = [v_{\pi(1)}, v_{\pi(2)}, ..., v_{\pi(|V|)}]$ is the core of DAG scheduling, where $\pi : \{1, 2, ..., |V|\} \to \{1, 2, ..., |V|\}$. Note that $Solution(G)$ is not necessarily the actual execution order of $G$. Rather, it is a topological order of $G$, while each valid $Solution(G)$ does correspond to one valid execution order. We regard each step in constructing this topological order as a decision point, while the selectable nodes at each step are define as the current **candidate nodes**. The ultimate goal is to generate a $Solution(G)$ that optimizes the target, such as makespan.

### 2.3 POINTER NETWORK FOR DAG SCHEDULING.

The studies that use the foundational PtrNet by Bello et al. (2016) for DAG scheduling (Dong et al. (2021); Zhao et al. (2022); Chen & Wang (2024); Li et al. (2022)) apply a LSTM decoder structure to obtain the current states' embeddings, as shown in Equation (1), where $x_1, x_2, ..., x_{|V|}$ are the feature vectors of each task node , $h^{(z)}$ is the encoder's output, and $h_t^{(s)}$ is the decoder's hidden state at $t$. They then calculate the attention scores $(u_i^t)$ among $h_t^{(s)}$ and the task nodes in the candidate node set $S_C^{(t)}$ by Equation (2), where $a$, $W_{ref}$ and $W_q$ are learnable matrices.

$$h^{(z)} = \text{LSTMEncoder}(x_1, x_2, ..., x_{|V|}), h_t^{(s)} = \text{LSTMdecoder}(h_{t-1}^{(s)}, h^{(z)}) \qquad (1)$$

$$u_i^t = \begin{cases} a^T \tanh(W_{ref} h_i + W_q h_t^{(s)}), i \in S_C^{(t)} \\ -\infty, i \notin S_C^{(t)} \end{cases} \tag{2}$$

As for those DAG scheduling studies following the PtrNet of Kool et al. (2018) (e.g., Lee et al. (2020; 2021) ), rather than using a LSTM-based structure, they construct a context embedding $h_t^{(c)}$ by concatenating the feature vectors of the task node that are chosen in recent decisions, and the global feature vector of $G$. They then calculate the attention scores at $t$ ($u_i^t$) of the candidates in $S_C^{(t)}$ according to $h_t^{(c)}$ by Equation (3), where $W_Q$, $W_K$ are learnable matrices and $dim$ is the embedding's dimension.

$$u_i^t = \begin{cases} \frac{(W_Q h_t^{(c)})(W_K h_i)^T}{\sqrt{dim}}, i \in S_C^{(t)} \\ -\infty, i \notin S_C^{(t)} \end{cases} \tag{3}$$

After getting the attention scores ($u_i^t$) of the nodes in $S_C^{(t)}$, these methods would apply a softmax in order to obtain the probability of selecting each task node among $S_C^{(t)}$, as shown in Equation (4), where $C_{clip}$ is the constant coefficient to clip the unnormalized log-probabilities.

$$P(v_i|t) = \frac{\exp(C_{clip} \cdot \tanh(u_i^t))}{\sum_{v_j \in V} \exp(C_{clip} \cdot \tanh(u_j^t))} \tag{4}$$

## 3 PROPOSED METHOD

### 3.1 MOTIVATION

Traditional PtrNet-based DAG scheduling methods rely on repeatedly computing the decoder hidden states or context embeddings according to the current observed environment state and the previous decisions at every decision step, and calculating the candidate node attention scores in order to obtain differentiable decision probabilities. Thus, these models tend to rely heavily on recent local decisions, at the expense of capturing the DAG's global structural properties. Furthermore, the repeated computation causes high computational complexity.

We believe that these limitations can be addressed by leveraging the graph attention and its aggregation. We define $S_F^{(t)}$ as the set of already scheduled task nodes at timestep $t$. Considering that the principle of PtrNet lies in computing the candidate node's attention scores according to the current state (represented by the decoder hidden states or context embeddings), such attention can be obtained directly by aggregating the graph attention scores between $S_F^{(t)}$ and $S_C^{(t)}$ (the candidate node set). Compared with the method in the existing PtrNets, graph attention is more effective in capturing the topological features of DAGs. Moreover, the attention scores between graph nodes can be obtained with one shot forward propagation of the network, which reduces the time complexity.

In this section, we propose GAA-PtrNet, a novel PtrNet based on GAA for one-shot DAG scheduling. GAA-PtrNet consists of a trainable network to calculate pair-wise node attention scores, and a scheduler that aggregates attention scores to obtain sampling probabilities. In the network, we utilize a GNN encoder to extract the DAG workflow to generate node embeddings, then we utilize graph attention mechanism to calculate pair-wise attention scores between all task nodes. The embeddings and attention scores are obtained through a one-shot network forward propagation, which has low time complexity. In the scheduler, at each sequencing step in scheduling, the sampling probabilities of each candidate nodes is obtained through GAA between the scheduled tasks and each candidate tasks. We also design a training strategy based on policy gradient RL. The overview of the proposed method is illustrated in Fig 2.

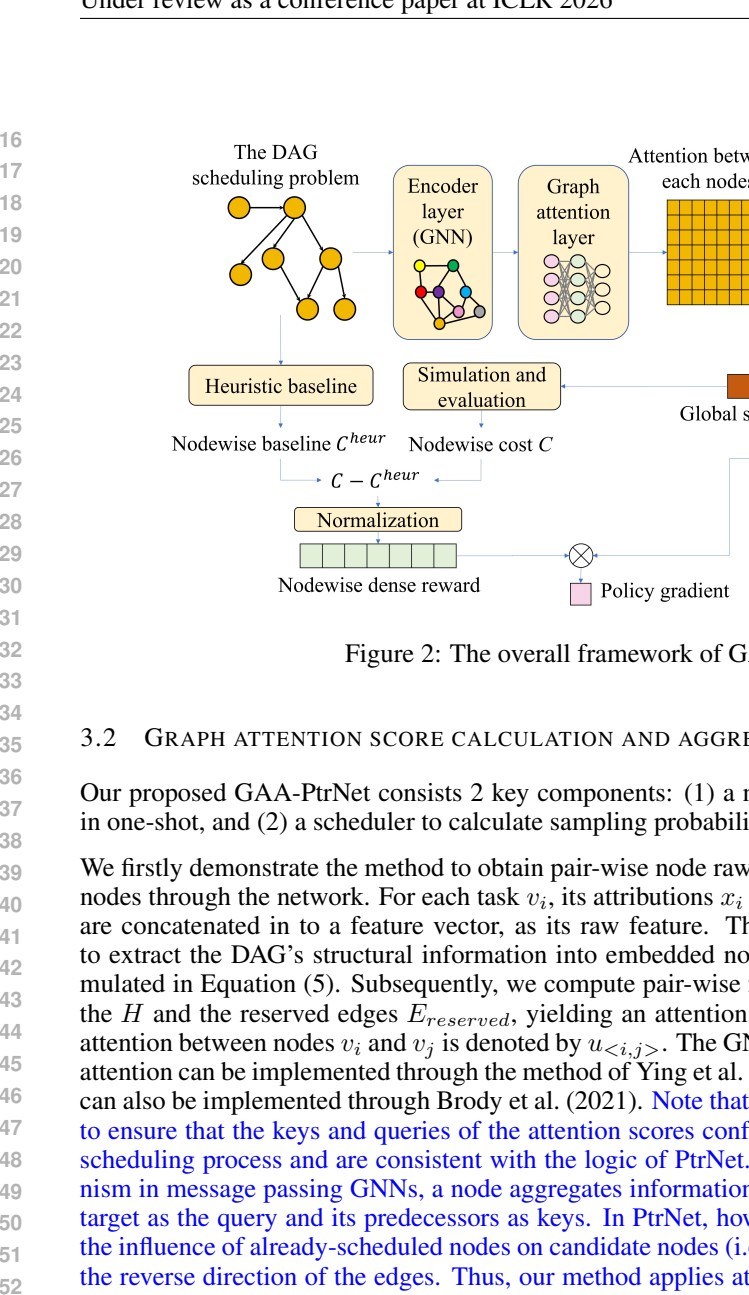

Figure 2: The overall framework of GAA-PtrNet.

## 3.2 GRAPH ATTENTION SCORE CALCULATION AND AGGREGATION

Our proposed GAA-PtrNet consists 2 key components: (1) a network to calculate attention scores in one-shot, and (2) a scheduler to calculate sampling probability through GAA.

We firstly demonstrate the method to obtain pair-wise node raw graph attention scores between task nodes through the network. For each task $v_i$, its attributions $x_i$ along with environment information are concatenated in to a feature vector, as its raw feature. Then we utilize a bi-directional GNN to extract the DAG's structural information into embedded node features $[h_1, h_2, ..., h_{|V|}]$, as formulated in Equation (5). Subsequently, we compute pair-wise node raw graph attention score from the $H$ and the reserved edges $E_{reserved}$, yielding an attention matrix $U$ (Equation (6)), where the attention between nodes $v_i$ and $v_j$ is denoted by $u_{<i,j>}$. The GNN and the pair-wise node raw graph attention can be implemented through the method of Ying et al. (2021). The pair-wise node attention can also be implemented through Brody et al. (2021). Note that in Equation (6), the edge is reserved to ensure that the keys and queries of the attention scores conform to the causal relationship in the scheduling process and are consistent with the logic of PtrNet. In standard graph attention mechanism in message passing GNNs, a node aggregates information from its predecessors—treating the target as the query and its predecessors as keys. In PtrNet, however, the attention should represent the influence of already-scheduled nodes on candidate nodes (i.e., scheduled → candidate), which is the reverse direction of the edges. Thus, our method applies attention on the reversed DAG to preserve consistency with graph attention formulations while ensuring the computed priorities reflect the correct causal flow.

$$H = [h_1, h_2, ..., h_{|V|}] = \text{GNN}(G) \tag{5}$$

$$U = [u_{<i,j>}] = \text{GraphAttn}(H, E_{reserved}) \tag{6}$$

Then, the scheduler computes the decision probability at each step directly according to the graph attention scores. The attention score $\alpha^{(t)}_{<i,j>}$ of attending $S^{(t)}_F$ to each candidate task node $v_j \in S^{(t)}_C$ is calculated by applying a softmax operation over the raw attention scores at the level of the full sets between $S^{(t)}_F$ and $S^{(t)}_C$, as shown in Equation (7).

$$\alpha^{(t)}_{<i,j>} = \frac{\exp(u_{<i,j>})}{\sum_{v_k \in S^{(t)}_C, v_l \in S^{(t)}_F} \exp(u_{<k,l>})}, v_i \in S^{(t)}_F, v_j \in S^{(t)}_C \tag{7}$$

For each candidate node $v_j \in S^{(t)}_C$, we aggregate its attention scores from $S^{(t)}_F$ by summation, as formulated in Equation (8). This yields $\alpha_{<S^{(t)}_F,j>}$, the attention from whole subgraph $S^{(t)}_F$ to $v_j$,

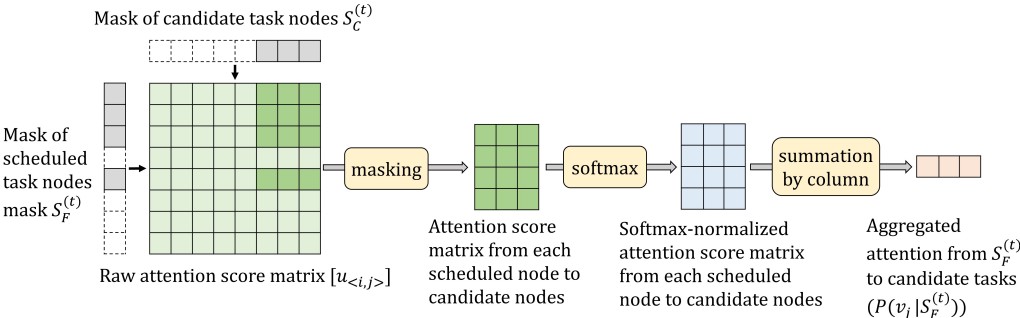

Figure 3: The procedure to obtain sampling probabilities of candidates by GAA. Each term in the raw attention score matrix indicates the attention from a task node (indexed by row) to another (indexed by column). In the attention masks (on the left of the figure), the dark cells indicate the scheduled or candidate nodes. These masks are applied to the raw attention score matrix to obtain the masked attention matrix at $t$. A softmax operation is then performed over the masked attention (Equation (7)), followed by a column-wise summation (Equation (8)), resulting in the aggregated normalized attention from $S_F^{(t)}$ to each in $S_C^{(t)}$, which is the probabilities of selecting each candidates.

which also corresponds to the probability $P(v_j|S_F^{(t)})$ of selecting $v_j$ from $S_C^{(t)}$ at time step $t$. This probability is differentiable and will be further used in the loss function computation. According to the probability distribution in Equation (8), we would sample the node $v_{\pi(t)}$ from $S_C^{(t)}$, as the decision at $t$. This procedure is illustrated in Fig 3. Note that as established in Section 3, we have assumed that each DAG workflow is augmented with a dummy entrance task node. The dummy entry node is introduced to ensure that the attention aggregation in each scheduling step can always operate on a non-empty set of already scheduled nodes. Without it, the first aggregation step would face an empty input set. At time step 0, when no nodes have been scheduled yet ($S_F^{(0)} = \emptyset$), we skip the aggregation and the directly select this dummy entrance node. By adding this virtual node, the GNN can also obtain meaningful information to initialize the scheduling process.

$$P(v_j|S_F^{(t)}) = \alpha_{<S_F^{(t)},j>} = \sum_{v_i \in S_F^{(t)}} \alpha_{<i,j>}^{(t)} \tag{8}$$

In Equation (8), we make a column-wise summation, which is also illustrated in Fig. 3. The column-wise summation is used to convert the attention matrix into a valid probability distribution over candidate nodes, making it consistent with the pointer mechanism in PtrNet-based schedulers. The softmax result $\alpha_{<i,j>}^{(t)}$ in Equation (7) can also be interpreted as the probability to select the node pair $< i, j >$ from all the node pairs that $i \in S_F^{(t)}$ and $j \in S_C^{(t)}$. These pairwise choices are mutually exclusive, so the probability of selecting candidate node $j$ is the probability of selecting all the node pairs from $S_F^{(t)}$ to $j$, i.e., Equation (8).

Equation (7) regards that all raw attention scores are in a unified scale, but the queries for these attention scores are from different task nodes in $S_F^{(t)}$. This raises a potential concern: could differences in the scale of the attention scores across different queries introduce bias in the probabilities computed by Equation (8). We believe this issue can be mitigated. In most graph attention mechanism, and the node embeddings $H$ are generated by the shared network weights, ensuring that the resulting attention scores lie within a comparable scale. However, since the attention scores are still computed in an independent query-wise manner in Brody et al. (2021), such scale mismatch still exists. In practice, we observed scale differences across rows of $U$ in early experiments. Thus, when using Brody et al. (2021) to compute the pair-wise node attentions, we apply a normalization to the attention scores originating from the same query node $v_i \in S_F^{(t)}$ before applying Equation (7). This ensures that all query-specific attention values lie on a comparable scale and prevents dominating attention values. After applying this normalization, training became more stable. Therefore, we adopted the normalization operation in our experiments. As for the pair-wise node attention, which

adopts a global self-attention (Ying et al., 2021), all attention scores are computed within a unified scale, and thus do not require additional normalization.

## 3.3 GAA-PTRNET-BASED DAG SCHEDULING PROCESS AND TIME COMPLEXITY ANALYSIS.

The scheduling procedure is demonstrated in Algorithm 1 in Appendix A. Given the scheduling problem $G$, the network firstly calculates $H$, then the scheduler updates $S_C^{(t)}$ and $S_F^{(t)}$ step by step, and repeatedly aggregates attention to obtain the sampling probabilities over $S_C^{(t)}$. At each step, a $v_{\pi(t)}$ is sampled accordingly, finally obtaining $Solution(G)$. An example about the procedure for GAA-PtrNet to schedule a DAG is also illustrated in Appendix B.

We analyze the time complexity for GAA-PtrNet to schedule $G = \{V, E, X\}$. Assume the embedding dimension in the network is $dim$, and the average size of $S_F^{(t)}$ and $S_C^{(t)}$ to be $\overline{|S_F|}$ and $\overline{|S_C|}$. An $L$-level GNN requires $O(L(|V| \times dim^2 + |E| \times dim))$ to generate node embeddings (take graph convolution network for example). Since different PtrNet-based scheduling methods may use similar GNN encoder, we mainly compare the time to make a complete schedule when the embeddings are already obtained. It takes $O(|E| \times dim + |V| \times dim^2)$ for Equation (12) or $O(|V|^2 + |V| \times |E| + |V|^2 \times dim + |V| \times dim^2)$ for Equation (11) to compute the raw attention scores. At each timestep, it takes $O(\overline{|S_F|} \times \overline{|S_C|})$ to conduct softmax, and probability calculation is conducted with with $O(\overline{|S_F|} + \overline{|S_C|})$ (upper bounded by $O(|V|)$). As a brief summary, the time complexity for GAA-PtrNet to make a schedule is $O(|E| \times dim + |V| \times dim^2 + |V| \times (\overline{|S_F|} \times \overline{|S_C|}))$ or $O(|V|^2 \times dim + |V| \times dim^2 + |V|(|V| + \overline{|S_F|} \times \overline{|S_C|}))$, which are both upper bounded by $O(|V|^2 \times dim + |V| \times dim^2 + |V|^3)$.

Table 1: Time complexity of GAA-PtrNet and existing PtrNet when scheduling DAGs.

| Method | Scheduling | Upper bound of scheduling |
|---|---|---|
| GAA-PtrNet | $O(|E| \times dim + |V| \times dim^2 + |V|(\overline{|S_F|} \times \overline{|S_C|}))$ | $O(|V|^2 \times dim + |V| \times dim^2 + |V|^3)$ |
| PtrNet-LSTM | $O(|V|^2 \times dim^2 + |V| \times \overline{|S_C|} \times dim^2 + |V| \times \overline{|S_C|})$ | $O(|V|^2 \times dim^2)$ |
| PtrNet-CE | $O(|V| \times \overline{|S_C|}^2 \times dim + |V| \times \overline{|S_C|} \times dim^2)$ | $O(|V|^2 \times dim^2 + |V|^3 \times dim)$ |

We denote the existing PtrNet base on LSTM structure and additive attention as **PtrNet-LSTM**, and the PtrNet structure with context embedding (CE) and dot-product attention as **PtrNet-CE**. Given the node embeddings, PtrNet-LSTM spends $O(|V|(dim^2 + \overline{|S_C|} \times dim^2))$ to generate a complete solution, in which $O(|V| \times dim^2)$ is the time for the LSTM cell in one step, and $O(\overline{|S_C|} \times dim^2)$ is additive attention. So the overall time complexity for PtrNet-LSTM is $O(|V|^2 \times dim^2 + |V||S_c| \times dim^2 + |V| \times |S_C|)$ upper bounded by $O(|V|^2 \times dim^2)$. Similarly, PtrNet-CE has a $O(|V| \times |S_C|^2 \times dim + |V||S_C| \times dim^2)$ time complexity, which is upper bounded by $O(|V|^2 \times dim^2 + |V|^3 \times dim)$, which is much higher than our proposed GAA-PtrNet. Moreover, in our method, the per-step computational complexity within the iterative scheduling loop is independent of the network dimension $dim$, as it involves only attention operations. Therefore, the advantage of our approach becomes more pronounced as the number of nodes in the DAG workflow increases. We intuitively compare the scheduling time complexity of GAA-PtrNet and existing PtrNet in Table 1.

## 3.4 TRAINING STRATEGY WITH POLICY GRADIENT RL

We use policy gradient to train our model. We apply dense reward signal for DAG scheduling, similar with the method proposed by Qi et al. (2025), in order to guide the optimization of each node-level decision. It is based on the distance of each node's contribution in the objective to the final objective. Please check Appendix G for more details about the implementation and analysis about this dense reward signal.

Demonstration learning is applied to initialize the training process, because the trajectories are sampled in a Monte Carlo way: it is generated in one-shot, and no greedy rule is applied in sampling. So, there might be blind exploration at the beginning of the training. We utilize genetic algorithm for DAG scheduling (Zhu et al., 2016b) to generate demonstration solutions to the scheduling problem

instances at first, obtaining some suboptimal solutions in the form of task execution orders. These solutions are converted to RL sampling trajectories and the network would be trained on these trajectories for several episodes. This can prevent the network from conducting inefficient exploration at the beginning of training.

## 4 EXPERIMENTS

### 4.1 EXPERIMENTAL SETUPS

In this section, we report experimental results to evaluate the contributions of our proposed method, and the performance of our method compared with existing method. The training and simulation of the experiments are conducted on a computer with Ubuntu 20.04 OS, Intel 6226R CPU, 256GB RAM, and RTX 3090 (24GB) graphic card. The experiments are conducted using Python 3.9 as the programming language. The neural network model is implemented based on PyTorch 2.7 and torch-geometric 2.6. Details about the implementation of experiments are presented in Appendix E.

In order to thoroughly evaluate the adaptability and robustness of our proposed method across different scenarios, we introduce the following benchmarks in our experiments: **TPC-H** represents the real-world DAG workflow scheduling scenario under homogeneous processor environment. TPC-H is collected from E-business oriented database query scenarios. It represents the typical computing task workflows in the E-business decision-making system. The DAG workflows in TPC-H are small but numerous. Here, we use the TPC-H benchmark generated by Wang et al. (2021). Each workflow in TPC-H has averagely 9.17 nodes. We tested on TPC-H with different workflow number (50, 100 and 150). **Pegasus** is a real-world scientific workflow scheduling tracing dataset with heterogeneous multiprocessor environment (Deelman et al., 2015). Pegasus workflows are collected from multiple scientific computing applications, including SIPHT, LIGO, GENOME, etc. Each type of application corresponds to a different workflow structure. The DAG workflows in Pegasus are large and complex. We tested on each type with different problem instance sizes (averagely 100, 200, 300, 400 and 1000 task nodes). **Randomly generated DAG workflows**, **TPC-H** and **Pegasus**. In the **randomly generated DAG workflows**, we generate DAG scheduling problems with varying shapes, sizes, and task node attributes by tuning the parameters, following the DAG generation paradigm of Topcuoglu et al. (2002). These parameters include graph shape parameter ($\beta_1$), task node heterogeneity ($\beta_2$), Computation-to-Communication Ratio ($CCR$) and average number of tasks in each sub-DAG. Please check Appendix D for more details about these benchmarks.

The target mainly includes average **makespan** of the DAG workflows. Also, **speedup** and **relative gap** are applied to evaluate the performance. Speedup indicates how many times the generated solution is faster in makespan to the case when all tasks are executed only on the fastest processor. Relative gap indicates the gap (in percentage) in makespan relative to the heuristic baseline. Besides, we test the **runtime** to infer a solution, in order to evaluate the time complexity of our method in practice. In the experiments, we train the model in parallel on 16 sampled problem instances as one batch, with up to 5000 episodes per batch.

### 4.2 RESULTS AND DISCUSSION

#### 4.2.1 ABLATION STUDY ABOUT GAA-PTRNET

We compare the performance of GAA-PtrNet on DAG scheduling to **PtrNet-LSTM** and **PtrNet-CE**. We also evaluate the influence of different attention mechanism used in GAA-PtrNet: the GAA-PtrNet implemented by self-product attention with position encoding (Ying et al., 2021), donated as **GAA-PtrNet-SA** , and the implementation based on classic graph attention network (Brody et al., 2021), donated as **GAA-PtrNet-GAT**. Note that these are originally for graph representation propose, not for scheduling, modifications are necessary, please check Appendix E.1 for more details.

As presented in in Table 2 (SIPHT), Table 3 (TPC-H), Table 5 in Appendix F.1 (LIGO), Table 6 (GENOME) in Appendix F.1 and Appendix F.2 (Randomly generated workflows), both **GAA-PtrNet-SA** and **GAA-PtrNet-GAT** outperform **PtrNet-CE** and **PtrNet-LSTM** in different evaluation matrices on diverse benchmarks. In most cases, the performance difference between **GAA-PtrNet-SA** and **GAA-PtrNet-GAT** is minor. This shows that the key factor behind the performance improvement is our proposed GAA, rather than which graph attention computation method is se-

lected. Due to page limits, please check Appendix F for more results. We evaluated the representative results of average runtime of each model, i.e., the time required for the model to infer a solution for a DAG scheduling instance (Table 4). To present these results visually, we have plotted the average runtime curves of each method on the each problem scale in Figure 4. The one-shot scheduling by GAA-PtrNet exhibits substantially better runtime performance than **PtrNet-LSTM** and **PtrNet-CE**. Our method averagely runs 10 time faster.

Table 2: Experimental results on SIPHT dataset in Pegasus.

| Method | SIPHT-100 | | | SIPHT-200 | | | SIPHT-300 | | | SIPHT-400 | | | SIPHT-1000 | | |
|---|---|---|---|---|---|---|---|---|---|---|---|---|---|---|---|
| | makespan | gap | speed up | makespan | gap | speed up | makespan | gap | speed up | makespan | gap | speed up | makespan | gap | speed up |
| GAA-PtrNet-SA | **191.1** | **-15.81** | **2.43** | 340.3 | -4.89 | 2.45 | 542.1 | -0.20 | 2.51 | 708.5 | -0.88 | 2.51 | 1818.8 | -0.14 | 2.51 |
| GAA-PtrNet-GAT | 191.7 | -15.55 | 2.42 | 338.8 | -3.41 | 2.51 | 542.7 | -0.05 | 2.50 | **708.3** | **-0.91** | **2.51** | **1818.6** | **-0.15** | **2.51** |
| PtrNet-LSTM | 205.0 | -9.69 | 2.27 | 352.8 | -1.40 | 2.41 | 552.2 | 1.69 | 2.46 | 717.5 | 0.38 | 2.48 | 1829.2 | 0.40 | 2.49 |
| PtrNet-CE | 207.5 | -8.59 | 2.24 | 354.8 | -0.84 | 2.40 | 557.0 | 2.58 | 2.44 | 719.6 | 0.67 | 2.476 | 1832.0 | 0.58 | 2.50 |
| HEFT (heuristic) | 227.0 | - | 2.05 | 357.8 | - | 2.38 | 543.0 | - | 2.51 | 714.8 | - | 2.49 | 1821.4 | - | 2.51 |
| Jeon et al. (2023) | 218.5 | -3.74 | 2.23 | 352.2 | -1.57 | 2.42 | 550.6 | 1.40 | 2.47 | 712.7 | -0.29 | 2.50 | 1898.1 | 4.21 | 2.40 |
| Qi et al. (2025) | 196.9 | -13.26 | 2.36 | 338.4 | -5.42 | 2.51 | 541.6 | -0.25 | 2.51 | 708.3 | -0.91 | 2.51 | 1819.2 | -0.13 | 2.51 |
| POMO-DAG | 214.0 | -5.74 | 2.17 | 367.5 | 2.72 | 2.31 | 575.8 | 6.05 | 2.36 | 741.9 | 3.80 | 2.40 | 1875.1 | 2.95 | 2.44 |
| EGS | 200.6 | -11.63 | 2.32 | 346.3 | -3.21 | 2.46 | 542.8 | -0.04 | 2.51 | 710.2 | -0.42 | 2.51 | 1821.0 | -0.02 | 2.51 |

Table 3: Experimental results on TPC-H benchmark, with 3 different problem instance sizes (50, 100 and 150 sub-DAGs, with each sub-DAG containing averagely 9.17 task nodes).

| Method | TPC-H 50 | | | TPC-H 100 | | | TPC-H 150 | | |
|---|---|---|---|---|---|---|---|---|---|
| | makespan | gap | speed up | makespan | gap | speed up | makespan | gap | speed up |
| GAA-PtrNet-SA | 21.37 | -14.42 | 5.25 | 39.59 | -7.54 | 5.37 | **67.08** | **-3.84** | 4.81 |
| GAA-PtrNet-GAT | 21.33 | -14.58 | 5.26 | 42.18 | -1.49 | 5.04 | 67.81 | -2.79 | 4.96 |
| PtrNet-LSTM | 26.10 | 4.53 | 4.30 | 44.94 | 4.95 | 4.73 | 73.20 | 4.93 | 4.41 |
| PtrNet-CE | 26.19 | 4.88 | 4.28 | 44.18 | 3.18 | 4.81 | 72.84 | 4.42 | 4.43 |
| STF (heuristic) | 24.97 | - | 4.49 | 42.85 | - | 4.96 | 69.76 | - | 4.63 |
| Jeon et al. (2023) | 23.73 | -4.95 | 4.73 | 41.22 | -3.81 | 5.15 | 74.02 | 6.11 | 4.35 |
| Qi et al. (2025) | **20.49** | **-17.73** | **5.47** | **39.22** | **-8.47** | **5.42** | 73.47 | 5.32 | 4.39 |
| POMO-DAG | 46.90 | 87.8 | 2.39 | 90.30 | 110.74 | 2.35 | 141.90 | 103.43 | 2.27 |
| EGS | 24.58 | -1.55 | 4.56 | 42.18 | -1.56 | 5.04 | 68.99 | -1.10 | 4.68 |

We further observed that the size $|V|$ is the dominant factor influencing runtime, while the choice of graph attention computation method has minor effect, which is consistent with the time complexity analysis. This demonstrates that GAA-PrtNet's runtime is less sensitive to the DAG topology. We believe this is because GAA-PrtNet must explicitly invoke $|V|$ times of loops to produce a complete solution, which is the most time-consuming operation. Moreover, it can be observed that the actual runtime of our GAA-PtrNet-based one-shot scheduling method increases relatively slowly with the growth of the scheduling problem size, unlike existing PtrNet models whose runtime scales linearly with problem size. We attribute this to the fact that conventional PtrNets repeatedly compute hidden states (or context embeddings) and attention at every scheduling decision, which accounts for the majority of the runtime, whereas our method avoids this overhead.

### 4.2.2 COMPARISON WITH BASELINE METHODS

Our approach is compared against these baselines: (1) The heuristic algorithms used as advantage baselines in our method; (2) Jeon et al. (2023), a RL-based one-shot DAG scheduling method based on performing list-scheduling on global logits list; (3) Qi et al. (2025), the newest RL-based one-shot DAG scheduler; (4) EGS (Sun et al., 2024), a DAG scheduling approach based on edge generation; (5) POMO-DAG, our adapted implementation of the POMO (Kwon et al., 2020) for DAG scheduling. Check Appendix E.3 for more details about the implementation of these baseline methods.

The results of comparison experiments are presented in Table 2 (SIPHT), Table 3 (TPC-H), Table 5 in Appendix F.1 (LIGO), Table 6 (GENOME) in Appendix F.1. In most cases, our method outperforms the RL-based one-shot scheduling method proposed by Jeon et al. (2023) and Qi et al. (2025). We argue that this is because their methods' reliance on performing list-scheduling on logits list caused higher variance in policy gradient and less stability in training. On the contrary, our method

does not need to conduct such ranking model. Under certain benchmark settings, the task nodes have smaller variance (e.g.,LIGO-400), or the task dependency paths are short (e.g., SIPHT-200), so the results of our method doesn't show a advantage over the logits list-based Qi et al. (2025). On the contrary, our method performs better in the other situations with diverse structures, such as most cases in Table 5 and Table 6. Since Jeon et al. (2023) and Qi et al. (2025) are one-shot scheduling methods, we also compared the runtime of their method with ours in Table 4 ind Figure 4. The results show that our method is consistently faster than Qi et al. (2025) in runtime, and also faster than Jeon et al. (2023) under the situation that there are more than 300 nodes. We attribute ours being faster than Qi et al. (2025) to that their comparability identification for the logits are more time-consuming. When compared with POMO-DAG, our method consistently yields better results. Compared to EGS, our method achieves similar or even better solution quality. We attribute this to that their search spaceis significantly larger. In conclusion, our method demonstrates an advantage in optimizing performance in most cases, especially for problems with more diverse structures, highlighting its stronger ability to handle with scheduling in complex high-performance computation scenarios. Meanwhile, our method also demonstrated faster runtime, providing advantages in scheduling for latency-sensitive applications.

Table 4: Runtime comparison on Pegasus of various scale (by second).

| Method | size = 100 | | | size = 200 | | | size = 300 | | | size = 400 | | | size = 1000 | | |
|---|---|---|---|---|---|---|---|---|---|---|---|---|---|---|---|
| | SIPHT | LIGO | GENOME | SIPHT | LIGO | GENOME | SIPHT | LIGO | GENOME | SIPHT | LIGO | GENOME | SIPHT | LIGO | GENOME |
| GAA-PtrNet-SA | 0.126 | 0.129 | 0.130 | 0.219 | 0.226 | 0.225 | 0.292 | 0.300 | 0.299 | 0.347 | 0.356 | 0.354 | 0.985 | 0.907 | 0.963 |
| GAA-PtrNet-GAT | 0.136 | 0.128 | 0.129 | 0.228 | 0.226 | 0.223 | 0.301 | 0.297 | 0.298 | 0.355 | 0.357 | 0.356 | 0.928 | 0.734 | 0.966 |
| PtrNet-LSTM | 1.343 | 1.389 | 1.387 | 2.645 | 2.718 | 2.710 | 3.921 | 4.054 | 4.015 | 5.202 | 5.391 | 3.994 | 12.478 | 12.881 | 12.912 |
| PtrNet-CE | 1.064 | 1.097 | 1.097 | 2.081 | 2.138 | 2.130 | 3.086 | 3.188 | 3.159 | 4.108 | 4.231 | 3.156 | 9.693 | 9.797 | 9.919 |
| Jeon et al. (2023) | 0.07 | 0.08 | 0.08 | 0.15 | 0.16 | 0.15 | 0.26 | 0.26 | 0.27 | 0.43 | 0.43 | 0.46 | 2.44 | 2.39 | 2.58 |
| Qi et al. (2025) | 0.61 | 0.62 | 0.61 | 0.97 | 0.99 | 1.00 | 1.17 | 1.20 | 1.20 | 1.22 | 1.26 | 1.29 | 2.59 | 2.59 | 2.79 |

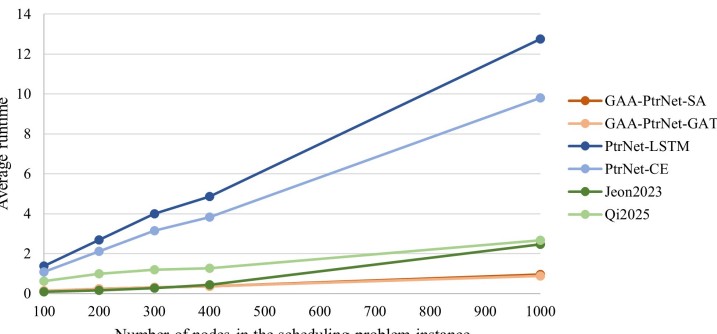

Figure 4: The average runtime comparison of each method on different problem scales (number of task nodes in the scheduling problem instance).

## 5 CONCLUSION

In this paper, we propose a novel GAA-PtrNet for DAG scheduling. By calculating the graph attention scores in a one-shot way and obtaining the task node sampling through GAA, GAA-PtrNet achieves one-shot DAG scheduling. It can handle with DAGs' complex topological structure with low time complexity. We also introduced a RL-based policy gradient training strategy for GAA-PtrNet. We conducted comparative and ablation experiments on various DAG scheduling scenarios, demonstrating the superiority of our method in solution quality and runtime. This suggest that our method have potential for further extension and optimization in large-scale or real-time DAG workflow environments. In future work, we will expand this foundational study to more specific high performance computation applications by considering domain knowledge and multiple Quality-of-service-based objectives, and expand our research to the promising area of real-time scheduling.

## 6 ETHICS AND REPRODUCIBILITY STATEMENT

We have read ICLR code of ethics and we claim no potential violations of the code of ethics. Regarding reproducibility, we describe the experimental environment in Section 4, provide details of the benchmark sources in Appendix D, and present the implementation details in Appendix E.

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

# A  THE ALGORITHM OF ONE-SHOT DAG SCHEDULING BY GAA-PTRNET

---

**Algorithm 1** Scheduling a DAG workflow instance by GAA-PtrNet

---

    **Input:** Workflow instance $G = (V, E, I)$
    **Output:** Priority topological sorting of ask node $[v_{\pi(1)}, v_{\pi(2)}, ..., v_{\pi(|V|)}]$
 1: Add a dummy entrance node $v_{dummy}$ to $G$
 2: Calculate the node embeddings $H$ by Equation (5)
 3: Calculate the raw attention scores $\{u_{<i,j>}\}$ by Equation (12) or Equation (11) on $G_{reserved}$.
 4: Set the time step $t = 1$, the priority sort $O = [v_{dummy}]$, the scheduled set $S_F^{(t)} = \{v_{dummy}\}$
 5: **while** $t \leq |V|$ **do**
 6:    Collet the current candidate task node set $S_C^{(t)}$
 7:    Softmax the raw attention scores over $S_C^{(t)}$ and $S_F^{(t)}$ by Equation (7)
 8:    Calculate the probability to select each node in $S_C^{(t)}$ by Equation (8)
 9:    Sample $v_{\pi(t)}$ based on the probabilities. Append $v_{\pi(t)}$ to $O$ and insert it to $S_F^{(t)}$
10:    Assign $v_{\pi(t)}$ to a processor according to a certain dispatching rule.
11: **end while**

---

# B  A ILLUSTRATIVE EXAMPLE OF ONE-SHOT DAG SCHEDULING BY GAA-PTRNET

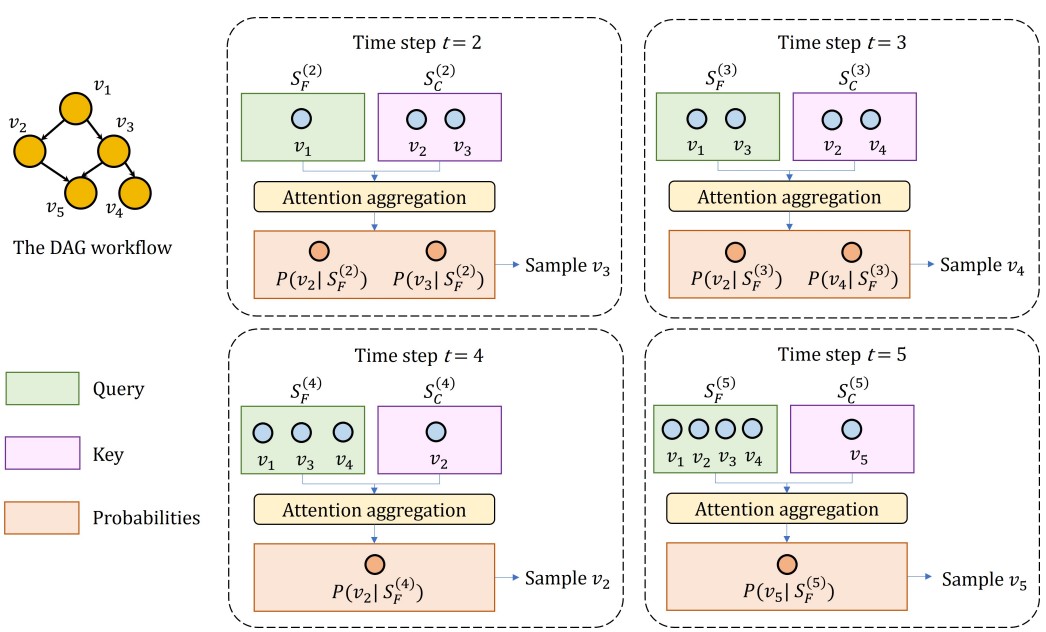

Figure 5: An example of scheduling a DAG with 5 task nodes by GAA-PtrNet

Fig 5 shows an example of scheduling a DAG with 5 task nodes by GAA-PtrNet. In Fig 5, at the beginning, the pseudo entry node $v_1$ is processed in advance. At timestep 2, the scheduled node subset is $S_F^{(2)} = v_1$, and the candidate subset is $S_C^{(2)} = v_2, v_3$. Through GAA, we obtain the probability to sample $v_2$ and $v_3$, and the node that the scheduler eventually sample is $v_3$. Next, at timestep 3, the scheduled node subset is $S_F^{(3)} = v_1, v_3$, and the candidate subset is $S_C^{(2)} = v_2, v_4$. Assume $v_4$ is sampled. Then, similarly, the scheduler samples $v_2$ at timestep 4, and $v_5$ at timestep 5. The final priority list is $Solution(G) = [v_1, v_3, v_4, v_2, v_5]$.

# C  RELATED WORKS

## C.1  RL FOR DAG SCHEDULING

While most existing studies on DAG workflow scheduling in cloud computing and cluster computing area still rely on heuristic (Topcuoglu et al., 2002; Djigal et al., 2021) or meta-heuristic algorithms (Xie et al., 2021; Qin et al., 2023) , learning-based approaches are gradually becoming the mainstream. Yang et al. (2019) was the first to introduce model-based reinforcement learning in the distributed and cloud-based system to schedule scientific workflow. Mao et al. (2019) first proposed to introduce GNN into RL model in order to extract structural features of the workflow instances to make better decisions. But it cannot handle heterogeneous computing resources and large scale workflows. A GNN encoder to extract workflows' structural information, and a policy network to output scheduling decisions, have been the common network architecture in the RL-based workflow scheduling studies(Zhou et al., 2022; Song et al., 2023; Qi et al., 2024). Wang et al. (2021) and Sun et al. (2024) attempted to learn to prioritize task nodes indirectly by modifying the DAG's topological structure, rather than directly generating task priority lists. But these approaches often suffered from an excessively large search space. A common challenge for RL-based workflow scheduling approaches is sparse reward in training. Some works (Chen et al., 2023; Wang et al., 2025; Nasuta et al., 2024) tried to overcome this challenge, but they were limited in specific application domains, rather than providing general solutions.

## C.2  RL FOR ONE-SHOT DAG SCHEDULING

These earlier DAG scheduling approaches follow Markov decision process and generate a solution step by step, requiring repeated extraction of global environment features and recalculation of schedules, which results in high computational overhead. The recent work of Jeon et al. (2023) addressed this issue by developing one-shot neural scheduler, which generates all the sub-decisions through a single forward pass of the network. This one-shot GNN+RL method relies on generating a global logits list for all task nodes by the policy network, in which the Gumbel top-k trick (Kool et al., 2019) is introduced to perturb the global logits list. The logits list is treated as the task priority list to derive a task execution ordering via list-scheduling heuristics. However, the scheduling method which achieves ranking by generating a global list of logits inherently has large policy gradient variance. Additionally, such list-scheduling based on global logits list suffers from the fact that multiple distinct permutations of the logits list may correspond to the same valid schedule. This many-to-one mapping biases the probability of policy sampling, so that policy gradient estimation is biased in the learning process, which makes the scheduler training more prone to local optima. Qi et al. (2025) proposed a comparable antichain identification mechanism, from the perspective of reducing redundant pairwise comparisons among logits during ranking, partly addressed this issue. However, this method still depends on generating a global logits list to rank the task nodes.

Different from prior studies, by achieving a one-shot PtrNet, our GAA-PtrNet fundamentally addresses the above limitations from the perspective of omitting the ranking on logits lists.

## C.3  RL COMBINED WITH PTRNET FOR DAG SCHEDULING

Just like other COP, workflow scheduling in high performance computing environment involves reordering the components (i.e., task , resource, or meta-heuristics rules) of the given problem instance. Under this background, PtrNet, as a sequence learning method, have shown its distinct advantages. PtrNet was initially proposed to handle sequence-to-sequence learning tasks in natural language processing area (Vinyals et al., 2015). Bello et al. (2016) firstly proposed to utilize PtrNet to solve routing problems, and Ma et al. (2019) introduced GNN to PtrNet as the encoder to better encode the graph-structured problem instance. Their basic idea was followed by many DAG workflow scheduling studies Dong et al. (2021); Zhao et al. (2022); Chen & Wang (2024); Li et al. (2022). To address the limitations of LSTM-based PtrNet in handling non-sequential structures, Deudon et al. (2018) modified the method of Bello et al. (2016) by computing the attention of context embedding, instead of the LSTM decoder's hidden state, to the candidate actions. In order to improve the computation parallelization and scaling potential, Kool et al. (2018) further modified the attention calculation in in PtrNet-based COP solver with dot-product attention (Vaswani et al., 2017), instead of the traditional additive attention mechanism (Bahdanau et al., 2014). Their method

are followed by many recent task scheduling studies Lee et al. (2020; 2021); Shi & Yu (2023); Wang et al. (2023). But in general, these PtrNet-based scheduling methods still suffer from limited capability of exploiting the global topological structure of DAGs, high time complexity and inability to achieve one-shot scheduling.

To our knowledge, our study is the first network model to achieve PtrNet-based one-shot DAG scheduling. It also achieves global topological information awareness for PtrNet model when scheduling DAGs.

## D  MORE INFORMATION ABOUT THE BENCHMARKS

### D.1  RANDOMLY GENERATED DAG WORKFLOWS

To evaluate the performance of our method in scheduling multiple workflow instances, we generate a set of random DAG workflows as test cases. We adopt it as the benchmark for testing DAG workflow scheduling algorithm under the simulated heterogeneous multiprocessor setting. Under such setting, the attributes of task node $v_i$ include the computational workload $c_i$ and the output data size $b_i$ . For each processor $m$, the key attribute is its computational capacity $f_m$. Assuming that task node $v_i$ is assigned to processor $m$, the processing time $d_i$ (as described in section 2) can be calculated as Equation Equation (9).

$$d_i = \frac{c_i}{f_m} \tag{9}$$

Besides, heterogeneous multiprocessor environment requires to consider the transmission time $z_i$ between processors: a task node cannot begin execution until all of its predecessor nodes have completed both their computation, and the transmission of their output data to the processor on which it is scheduled. Specifically, if a task node $v_i$ and its successor $v_j$ are assigned to different processors, then a transmission time is related to the output data size of $v_i$ and the bandwidth of computing environment. If both $v_i$ and $v_j$ are assigned to the same processor, the transmission time is considered negligible. This definition of $z_i$ is formalized in Equation Equation (10).

$$z_i = \begin{cases} \frac{b_i}{\text{bandwidth}}, & \text{if } v_i \text{ and } v_j \text{ are on different processors} \\ 0, & \text{if } v_i \text{ and } v_j \text{ are on the same processor} \end{cases} \tag{10}$$

Following Topcuoglu et al. (2002), the randomly generation process considers the following key parameters: (1) **Graph shape parameter** ($\beta_1$): this parameter characterizes the depth of the DAG. (2) **Task node heterogeneity** ($\beta_2$) : this captures the diversity in computational workloads $c_i$ and data sizes $b_i$ among different task nodes. (3) **Computation-to-Communication Ratio** ($CCR$): defined as the ratio between the average task processing time and the average data transmission time. (4) **Average number of tasks** in each sub-DAG in the whole DAG scheduling problem.

By tuning these parameters, we construct a diverse set of DAG workflows with varying structures and heterogeneity levels, simulating real-world heterogeneous environments in the parallel computing area. In the experimental setup, we vary one parameter at a time while fixing the remaining parameters to randomly assigned constant values. For each configuration, we generate multiple scheduling problem instances to examine how the method's performance changes with respect to the chosen parameter. Specifically, the parameter $\beta_1$ is tested with a range of $\beta_1 = \{0.1, 1.0, 1.5\}$. For $\beta_2$, it's tested with a range of $\beta_2 = \{1.0, 1.5\}$. $CCR$ is in a range of $CCR = \{0.1, 0.5, 1.0, 2.0, 5.0, 10.0\}$. $|V|$ is tested in a range of $|V| = \{10, 20, 30\}$. Each scheduling problem instance contains 20 independent sub-DAGs.

Our proposed method outputs only the execution order of the nodes, while the assignment of each node to a processor is determined using the Earliest Finish Time (EFT)-greedy rule. Specifically, for a given node to be scheduled, which is determined by the RL network, the EFT-greedy rule dispatches it to the processor that results in the earliest possible finish time. We adopt HEFT (Topcuoglu et al., 2002) as the advantage baseline for dense reward signal under this setting. HEFT is a classic list-based heuristic algorithm for DAG scheduling on heterogeneous processors. It computes the priority (rank-up) of each node based on the average finish time of its successor nodes, and then assigns each node to a processor using the EFT-greedy rule.

### D.2 TPC-H

We adopt TPC-H as the benchmark for DAG workflow scheduling under the homogeneous single-processor setting. Since it's a homogeneous processor scenario, there is no need to dispatch tasks to specific processors in such setting. Each task node $v_i$ has a fixed processing time $d_i$ and computation resource requirement $q_i$. The total resource consumption of concurrently running tasks must not exceed the system's maximum resource capacity. We use the open source code [1] implemented by Wang et al. (2021) to generate TPC-H instances. Shortest Time First (STF) heuristic is adopted as the advantage baseline for TPC-H in our research. At each decision point, the STF rule selects the task with the shortest processing time.

### D.3 PEGASUS

Pegasus (Deelman et al., 2015) [2] provides an open-source workflow trace data generated from various scientific computing applications. We adopt LIGO, SIPHT and Genome data in Pegasus dataset as the benchmark for DAG workflow scheduling under the real-world heterogeneous multiprocessor setting. Since it's also a heterogeneous multiprocessor environment, similar with the randomly generated workflows, the task processing time is calculated by Equation (9) and transmission time should be considered calculated by Equation (10). EFT-greedy rule is utilized to dispatch the scheduled task nodes to the processors. HEFT is used as the advantage baseline for the dense reward signal.

## E IMPLEMENTATION DETAILS

### E.1 IMPLEMENTATION ABOUT THE GRAPH ATTENTION FOR SCHEDULING

We compare GAA-PtrNet with 2 different graph attention calculation methods: the GAA-PtrNet implemented by self-product attention with position encoding in Equation (11) (Brody et al., 2021) , and the implementation by classic graph attention in Equation (12) (Ying et al., 2021), where $a$, $W$, $W_Q$ and $W_K$ are learnable weight matrices, and $dim$ is the embedding dimensionality. Equation (11) applies self-attention over all node pairs at the graph level, and incorporates a learnable bias term $b_{\phi(v_i,v_j)}$ to encode the shortest-path distance $\phi(v_i, v_j)$ between nodes. Equation (12) computes attention only between adjacent nodes, and we mask out non-adjacent pairs.

$$u_{<i,j>} = [\frac{(HW_Q)(HW_K)^T}{\sqrt{dim}}]_{<i,j>} + b_{\phi(v_i,v_j)} \tag{11}$$

$$u_{<i,j>} = \begin{cases} a^T W[h_i||h_j] & , (v_j, v_i) \in E \\ -\infty & , (v_j, v_i) \notin E \end{cases} \tag{12}$$

It is important to note that in the original formulation of Brody et al. (2021), a LeakyReLU layer is applied to $u_{<i,j>}$, because it is then used to compute weights for the following feature aggregation. However, in our method, since we directly use the raw attention scores to compute softmax probabilities, we omit the activation layer in Equation (12). This modification ensures that the softmax is directly conducted on the raw attention logits, and the results are more interpretable as probability.

Moreover, in the original paper of Brody et al. (2021) and Ying et al. (2021), the attention follows the direction of edges in the graph. But in GAA-PtrNet, the attention scores is employed to compute the sampling probabilities. Therefore, to compute the attention from the scheduled node set to the unscheduled nodes, the edge directions must be reversed. As a result, we apply these attention module to the reversed DAG, $G_{reserved}$.

---

[1] https://github.com/Thinklab-SJTU/PPO-BiHyb/tree/main/dag_data/tpch
[2] https://pegasus.isi.edu/workflow_gallery/

### E.2 Neural network and hyper parameter settings

In the trainable neural network of GAA-PtrNet, as well as the implementation of **PtrNet-LSTM** and **PtrNet-CE** in ablation study, we use a Graphormer(Ying et al., 2021) GNN [3] (which is released under the MIT license) with 4 layers and 4 attention heads, obtaining node embeddings $[h_1, h_2, ..., h_{|V|}]$. It processes both the input DAG and its reversed graph simultaneously and outputs 32-dimension node embeddings for each task nodes. In our proposed method and the ablation study, each attention mechanism used to output the raw attention scores for GAA is restricted to a single head, and the dimension of its learnable matrices is 32, in order to keep consistent with the dimension of the node embeddings. We train the model in parallel on 16 sampled problem instances as one batch, with up to 5000 episodes per batch, although it actually converged much earlier than this epoch. The Adam optimizer is employed with learning rate $5 \times 10^{-4}$. We set the genetic algorithm parameters as: the size of population = problem size * 2, crossover rate = 0.15, mutation rate=0.30, generation number = 200.

### E.3 Baseline implementation

**Jeon et al. (2023).** The original authors didn't release their source code, the idea described in their paper is sufficiently clear and straightforward for us to reproduce.

**Qi et al. (2025).** The recently published RL-based one-shot scheduling method. We reproduced their method.

**POMO-DAG.** We build POMO-DAG upon the POMO [4] proposed by Kwon et al. (2020), adapting its problem instance encoder to a Graphormer-based GNN (Ying et al., 2021) so that it can process DAG scheduling problems.

**EGS.** For EGS (Sun et al., 2024), the basic framework of the original code is publicly available[5]. We retained the original structure and implemented the missing policy network and training procedure that were not released.

### E.4 Simulation and evaluation environment

To evaluate the generated scheduling solutions and obtain both the overall optimization objective and node-level dense reward signals, we implemented a DAG workflow simulation environment based on the open-source SimPy [6] platform in Python language. This simulator is further wrapped into an OpenAI Gym[7] environment to integrate with reinforcement learning frameworks. It is capable to simulate all three aforementioned benchmarks, and can be extended to support other scheduling scenarios if necessary.

## F Additional experimental results

### F.1 Additional results on Pegasus benchmark

Table 5 and 6 presents the results on LIGO and GENOME of Pegasus benchmark. The DAG workflows in these benchmarks have relatively longer interdependence path and more complex structure than SIPHT. As shown in Table 5 and 6, our approach yields better results than traditional PtrNet-based scheduler, e.g.,PtrNet-LSTM and PtrNet-CE. Generally, our method also perform better than other learning-based DAG scheduling methods in most cases, which highlights our methods' stronger ability to handle with scheduling in complex high-performance computation scenarios.

---

[3] https://github.com/microsoft/Graphormer
[4] https://github.com/yd-kwon/POMO
[5] https://github.com/binqi-sun/egs
[6] https://simpy.readthedocs.io/en/
[7] https://github.com/openai/gym

Table 5: Experimental results on LIGO, Pegasus benchmark.

| Method | LIGO-100 | | | LIGO-200 | | | LIGO-300 | | | LIGO-400 | | | LIGO-1000 | | |
|---|---|---|---|---|---|---|---|---|---|---|---|---|---|---|---|
| | makespan | gap | speed up | makespan | gap | speed up | makespan | gap | speed up | makespan | gap | speed up | makespan | gap | speed up |
| GAA-PtrNet-SA | **211.0** | **-2.98** | **2.49** | 460.7 | -0.48 | 2.49 | 666.9 | -0.51 | 2.50 | 956.9 | -0.25 | 2.50 | 2374.7 | 0.05 | 2.50 |
| GAA-PtrNet-GAT | 211.8 | -2.62 | 2.48 | **460.6** | **-0.50** | **2.50** | **666.1** | **-0.63** | **2.51** | **956.6** | **-0.28** | 2.50 | 2374.2 | 0.03 | 2.50 |
| PtrNet-LSTM | 216.0 | -0.70 | 2.44 | 466.6 | 0.81 | 2.47 | 670.7 | 0.06 | 2.49 | 960.9 | 0.17 | 2.49 | 2375.5 | 0.08 | 2.50 |
| PtrNet-CE | 216.4 | -0.51 | 2.23 | 467.1 | 0.89 | 2.47 | 671.0 | 0.10 | 2.49 | 961.3 | 0.21 | 2.49 | 2394.7 | 0.89 | 2.48 |
| HEFT (heuristic) | 217.5 | - | 2.42 | 462.9 | - | 2.49 | 670.3 | - | 2.49 | 959.3 | - | 2.49 | 2373.5 | - | 2.50 |
| Jeon et al. (2023) | 217.0 | -0.23 | 2.42 | 465.1 | 0.47 | 2.48 | 670.8 | 0.07 | 2.49 | 962.8 | 0.36 | 2.49 | 2376.7 | 0.13 | 2.50 |
| Qi et al. (2025) | 214.0 | -1.61 | 2.46 | 462.1 | -0.17 | 2.50 | 668.8 | -0.22 | 2.49 | **956.6** | **-0.28** | 2.50 | **2373.5** | **0** | **2.50** |
| POMO-DAG | 216.7 | -0.37 | 2.43 | 473.8 | 2.35 | 2.43 | 675.3 | 0.75 | 2.47 | 969.1 | 1.02 | 2.47 | 2382.0 | 0.36 | 2.50 |
| EGS | 214.2 | -1.52 | 2.46 | 462.5 | -0.09 | 2.49 | 669.9 | -0.06 | 2.49 | 956.9 | -0.25 | 2.50 | **2373.5** | **0** | **2.50** |

Table 6: Experimental results on GENOME, Pegasus benchmark.

| Method | GENOME-100 | | | GENOME-200 | | | GENOME-300 | | | GENOME-400 | | | GENOME-1000 | | |
|---|---|---|---|---|---|---|---|---|---|---|---|---|---|---|---|
| | makespan | gap | speed up | makespan | gap | speed up | makespan | gap | speed up | makespan | gap | speed up | makespan | gap | speed up |
| GAA-PtrNet-SA | **2435.5** | **-4.61** | 2.44 | 2351.2 | -0.94 | 2.46 | 4723.3 | -0.60 | 2.48 | **3451.1** | **-0.06** | 2.48 | **14948.1** | **-0.46** | **2.50** |
| GAA-PtrNet-GAT | 2438.4 | -4.49 | 2.44 | **2348.1** | **-1.07** | **2.47** | **4721.4** | **-0.64** | 2.48 | 3473.5 | 0.559 | 2.47 | 14966.2 | -0.34 | 2.49 |
| PtrNet-LSTM | 2497.4 | -2.18 | 2.38 | 2386.9 | 0.57 | 2.43 | 4788.4 | 0.77 | 2.45 | 3505.5 | 1.51 | 2.44 | 14982.2 | -0.20 | 2.49 |
| PtrNet-CE | 2493.9 | -2.32 | 2.38 | 2386.2 | -0.54 | 2.44 | 4796.1 | 0.93 | 2.44 | 3500.6 | 1.39 | 2.45 | 15007.9 | -0.06 | 2.49 |
| HEFT (heuristic) | 2553.1 | - | 2.33 | 2373.4 | - | 2.44 | 4751.9 | - | 2.46 | 3453.2 | - | 2.48 | 15016.8 | - | 2.48 |
| Jeon et al. (2023) | 2511.4 | -1.63 | 2.37 | 2369.9 | -0.15 | 2.48 | 4755.3 | 0.07 | 2.46 | 3483.2 | 0.87 | 2.46 | 15001.9 | -0.10 | 2.49 |
| Qi et al. (2025) | 2468.2 | -3.32 | 2.41 | 2350.5 | -0.96 | 2.47 | 4728.4 | -0.49 | 2.48 | 3453.0 | -0.01 | 2.48 | 14955.8 | -0.41 | 2.50 |
| POMO-DAG | 2472.0 | -3.18 | 2.41 | 2367.2 | -0.26 | 2.45 | 4783.2 | 0.66 | 2.45 | 3527.6 | 2.16 | 2.43 | 15005.7 | -0.07 | 2.49 |
| EGS | 2475.6 | -3.04 | 2.40 | 2356.2 | -0.72 | 2.46 | 4730.0 | -0.46 | 2.48 | 3453.0 | -0.01 | 2.48 | 14970.4 | -0.31 | 2.49 |

We have also conducted experiments on MONTAGE and CYBERSHAKE. However, due to the small variance in computational workload among task nodes in these workflows, different methods produces almost indistinguishable results across different methods under the fundamental DAG scheduling setting. For this reason, we only briefly report the makespan here in Table 7.

Table 7: Additional experimental results (makespan) on MONTAGE and CYBERSHAKE.

| Method | MONTAGE | | | | | CYBERSHAKE | | | | |
|---|---|---|---|---|---|---|---|---|---|---|
| | 100 | 200 | 300 | 400 | 1000 | 100 | 200 | 300 | 400 | 1000 |
| GAA-PtrNet-SA | 13.8 | 26.8 | 40.3 | 54.4 | 135.1 | 28.2 | 50.1 | 76.3 | 100.5 | 252.7 |
| GAA-PtrNet-GAT | 13.8 | 27.0 | 40.3 | 54.4 | 135.2 | 28.2 | 50.2 | 76.4 | 100.6 | 253.0 |
| PtrNet-LSTM | 13.9 | 26.8 | 40.6 | 54.4 | 135.1 | 28.2 | 50.3 | 76.5 | 100.5 | 252.7 |
| PtrNet-CE | 14.0 | 26.8 | 40.6 | 54.4 | 135.1 | 28.3 | 50.3 | 76.5 | 100.6 | 253.0 |
| HEFT (heuristic) | 13.9 | 27.0 | 40.5 | 54.6 | 135.3 | 28.6 | 50.9 | 77.4 | 101.6 | 255.0 |
| Jeon et al. (2023) | 13.8 | 26.9 | 40.5 | 54.4 | 135.1 | 28.3 | 50.2 | 76.3 | 100.6 | 252.9 |
| Qi et al. (2025) | 13.9 | 26.9 | 40.4 | 54.4 | 135.1 | 28.3 | 50.1 | 76.3 | 100.7 | 252.9 |

## F.2 ADDITIONAL RESULTS ON RANDOM GENERATED DAG WORKFLOWS

On the randomly generated DAG workflows, the performance differences between methods become more pronounced. Therefore, we present the results using bar charts. In each chart in Figure 6, 7, 8 and 9, we show the outcomes when varying a single DAG generation parameter. The x-axis denoting the parameter values and each bar representing a different method. Since the makespan varies greatly across different parameter settings, the y-axis reports the speedup instead of the makespan.

It can be noticed that, although the difference is minor in most cases, in some special cases of randomly generated DAGs, the GAA-PtrNet-SA-based scheduling method obviously outperforms GAA-PtrNet-GAT. We attribute this to its attention computation mechanism (originally proposed by (Ying et al., 2021)), which does not limit attention to adjacent edges but instead leverages self-attention combined with positional encoding to globally compute attention among all nodes in the scheduling problem. This enables the model to capture dependencies even between nodes belonging to disjoint sub-DAGs, thereby enhancing its performance in such cases.

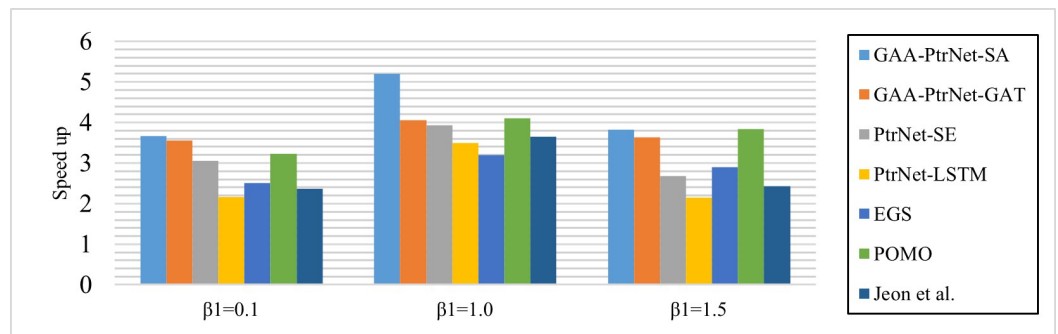

Figure 6: The results when adjusting parameter $\beta_1$.

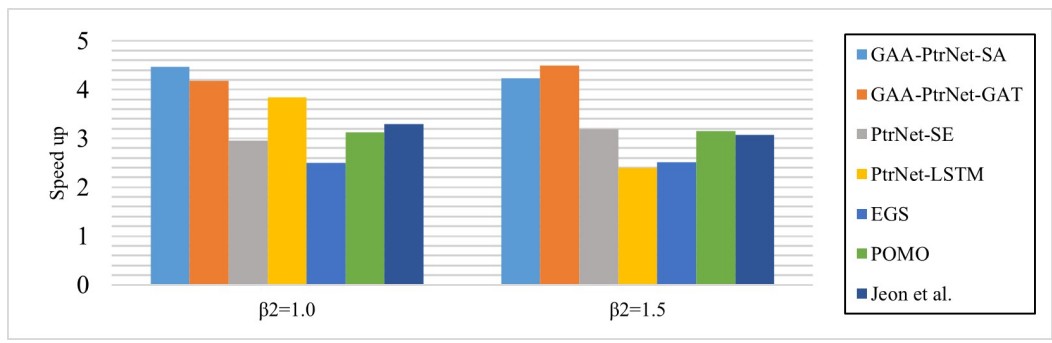

Figure 7: The results when adjusting parameter $\beta_2$.

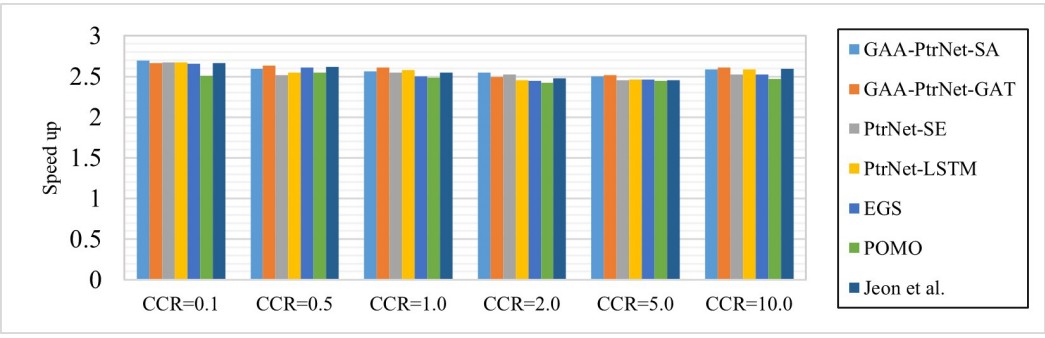

Figure 8: The results when adjusting parameter $CCR$.

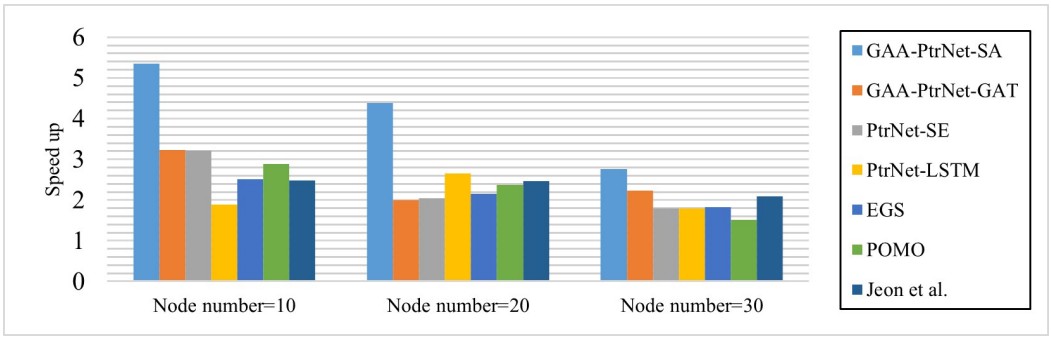

Figure 9: The results when adjusting the number of nodes in each sub-DAG.

### F.3 ADDITIONAL EXPERIMENTAL RESULTS ON JOB SHOP SCHEDULING PROBLEMS (JSSP).

We conducted additional experiment on job shop scheduling problems (JSSP), because JSSP is often regarded as a special case of DAG scheduling. We conducted experiments on 4 different scales: 20 jobs with 10 operations (20*10), 20 jobs with 20 operations (20*20), 30 jobs with 10 operations (30*10) and 30 jobs with 30 operations (30*20). The problem instances are randomly generated following the implementation of `https://github.com/zcaicaros/L2D/blob/main/DataGen/`. The results are reported in Table 8.The results show that our method clearly outperforms the baselines. This is because the baselines rely solely on a global priority list over operations. When applying a global priority list, many operations have to wait for higher-priority ones on the same machine, leading to much longer machine idle time. In contrast, our method dynamically derives the optimal scheduling decision for each idle machine at every decision point through attention aggregation, thereby avoiding the above issue.

Table 8: Experimental results of makespan on JSSP.

| Method | JSSP 20*10 | JSSP 20*20 | JSSP 30*10 | JSSP 30*20 |
|---|---|---|---|---|
| **GAA-PtrNet-SA** | 181.9 | 283.4 | 275.8 | 387.0 |
| **GAA-PtrNet-GAT** | 194.3 | 284.7 | 269.7 | 391.8 |
| **PtrNet-LSTM** | 217.4 | 313.3 | 321.3 | 428.1 |
| **PtrNet-CE** | 217.5 | 303.4 | 286.1 | 433.0 |
| SPT (Baseline) | 516.7 | 1096.2 | 845.9 | 1692.0 |
| Jeon et al. (2023) | 445.2 | 964.0 | 735.6 | 1548.6 |
| Qi et al. (2025) | 397.3 | 813.8 | 571.3 | 1426.0 |
| POMO-DAG | 341.6 | 936.3 | 971.3 | 1458.0 |
| EGS | 465.9 | 1034.5 | 837.5 | 1604.1 |

### F.4 ABLATIONS ABOUT THE COLUMN-WISE NORMALIZATION ON THE PAIR-WISE NODE ATTENTIONS

Under the experiment settings in Appendix E, we conducted some ablation about the column-wise normalization on the pair-wise node attentions $U$ before applying Equation (7) when using **GAA-PtrNet-GAT**, which is mentioned in Section 3.2, in order to evaluate its influence. The results is presented in Figure 10. The results show that the normalization of the attention scores can lead to faster converge.

## G DESCRIPTION AND ANALYSIS OF THE DENSE REWARD SIGNAL

Following Qi et al. (2025), the node-level reward signal for each scheduled node is computed based on its distance to the final cost. The return $R_{(t)}$ at time step $t$ can be estimated by Equation (13), where $C(Solution(G))$ is the final makespan value of $Solution(G)$, and $C(v_{\pi(t)})$ indicates the latest completion time among all scheduled nodes when $v_{\pi(t)}$ is finished. The value of $C(v_{\pi(t)})$ can be obtained during the simulation procedure when computing $Solution(G)$. By introducing an advantage baseline as shown in Equation (14), the advantage function $A_{(t)}$ can be cauculated as shown in Equation (15).

$$R_{(t)} = C(Solution(G)) - C(v_{\pi(t)}) \tag{13}$$

$A_{(t)}$ can be further normalized batch-wise into $A_{(t)}^{\text{normalized}}$ when there are multiple workflow instances $\mathcal{G} = G_1, G_2, ..., G_B$ in training. In this way, the policy gradient loss function can be modified into Equation (16), where $B$ is the batch size, and $\theta$ represents all the parameters in the trainable neural network.

$$\textbf{baseline} = C(Solution(G)) - C^{heur}(v_{\pi(t)}) \tag{14}$$

$$A_{(t)} = R_{(t)} - \textbf{baseline} = C^{heur}(v_{\pi(t)}) - C(v_{\pi(t)}) \tag{15}$$

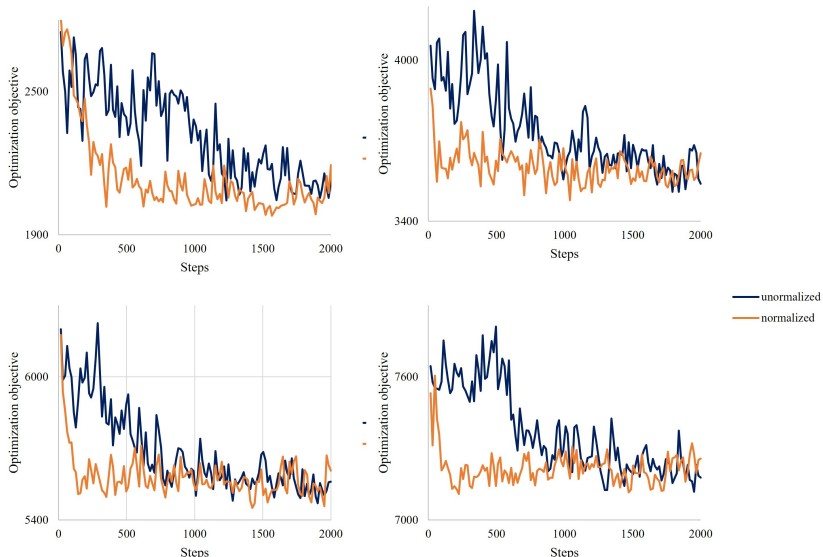

Figure 10: The curves that show the variation of the optimization objective (makespan) with the number of iterations. The four pictures respectively represent one of the sampled workflow respectively in SIPHT-100, 200, 300 and 400.

$$\nabla_\theta J(\theta) = \frac{1}{B} \frac{1}{|V_m|} \sum_{j=1}^{B} \sum_{t=1}^{|V_m|} \nabla_\theta \log P(v_{\pi(t)}|S_F^{(t)}) \cdot A_{m,(t)}^{\text{normalized}} \tag{16}$$

Specifically:

1. Given a DAG scheduling problem $G$, we conduct the heuristic advantage baseline algorithm on $G$, obtaining each task node's individual baseline cost $C^{heur}(v_{\pi(t)})$.

2. For a sampled solution $Solution(G)$, we simulate it using a SimPy-based simulator. and obtain the overall makespan $C(Solution(G))$ in practice. For each task node $v_{\pi(t)}$, we obtain its individual cost $C(v_{\pi(t)})$ (e.g., its finish time) through simulation.

3. We obtain the return-like dense reward signal $R_t$ of each task node $v_{\pi(t)}$ by comparing the global objective $C(Solution(G))$ with individual cost $C(v_{\pi(t)})$, according to Equation Equation (13).

4. To derive the advantage-like feedback $A_t$, we further substract each baseline from $R_t$, according to Equation Equation (15).

Figure 11 shows the curves of the makespan values evaluated on SIPHT-100 and SIPHT-200 during training of GAA-PtrNet, with and without dense reward signals, as a function of training epochs. It can be observed that introducing dense reward signals does not affect the quality of the final convergence, but instead leads to faster and more stable convergence.

We found that the selection of advantage baseline used in the dense reward is minor. This is because the heuristic algorithm provide constant estimates for each DAG scheduling problem instance, ensuring the advantage estimation is unbiased. Additionally, these heuristics are near-optimal in many cases, leading to similar schedules. As a result, the variance reduction benefit is preserved, while introducing little bias.

By introducing the dense reward signal, our method improves the interpretability of one-shot scheduling approaches to some extent. Specifically, the nodewise decisions, sampling probabilities and dense reward signals can be regarded as an entire MDP sampling trajectory in RL (like a trajectory by Monte Carlo sampling). This makes the interpretability of one-shot learning closer to MDP-based incremental approaches than those one-shot methods with sparse rewards.

As for the demonstration learning, it serves as the approach for the initialization of the training. Without demonstrations, we observe that the model may fall into blind exploration and even fail to converge at all. We used the genetic algorithm for DAG scheduling proposed by Zhu et al. (2016a) to generate demonstrations.

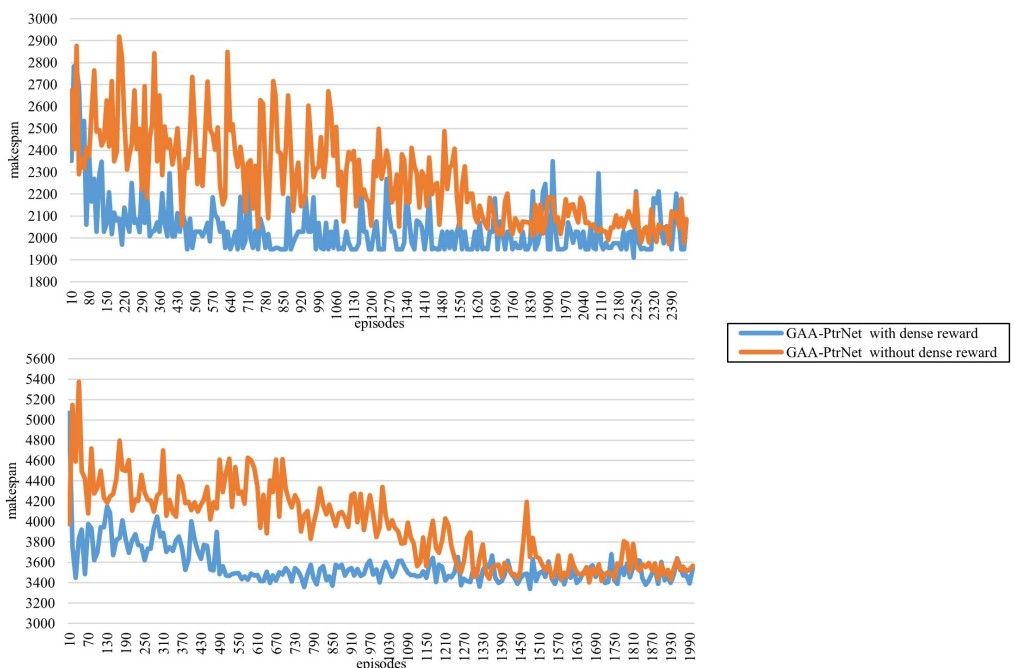

Figure 11: The curves of the makespan values evaluated on SIPHT-100 (the upper figure) and SIPHT-200 (the lower figure) during training of GAA-PtrNet, with and without dense reward signals.

## H  LLM USAGE IN THIS PAPER

In this paper, large language models are used only for writing embellishment and polishing, mainly focusing on certain sentences in the introduction and abstract. All the innovative points are original and no LLM was used for this propose.

