# OpenReview forum: "GAA-PtrNet: Graph attention aggregation-based pointer network for one-shot DAG scheduling"
_ICLR.cc/2026/Conference — Submitted to ICLR 2026_

### Official Review · Reviewer_vDpe · 2025-10-30

**Soundness:** 3
**Presentation:** 3
**Contribution:** 3
**Rating:** 8
**Confidence:** 4

**Summary:**

The authors proposed a Graph Attention Aggregation-based Pointer Network (GAA-PtrNet) for the Directed Acyclic Graph (DAG) scheduling problem to address the issues of traditional PtrNet in DAG scheduling, such as reliance on local decisions, high computational complexity, and inability to generate a scheduling plan in one shot. The model employs graph attention to compute the attention scores between nodes in one-shot, followed by an aggregation operation to obtain the probability of selecting candidate nodes. In addition, the authors designed a training strategy based on policy gradient reinforcement learning (RL). Through ablation and comparative experiments, the paper demonstrates the effectiveness of the proposed model architecture and its superiority in solving the DAG scheduling problem. In addition, this method is nearly 10 times faster than the traditional PtrNet.

**Strengths:**

1. The authors innovatively use graph attention to compute attention scores between graph nodes in a one-shot manner, thereby addressing the high computational complexity caused by repeated calculations in traditional PtrNet-based DAG scheduling methods and improving computational efficiency.

2. Through extensive comparative experiments, the authors compare their approach with existing heuristic methods and state-of-the-art reinforcement learning methods for DAG scheduling, verifying the superiority of the proposed method in both solution quality and computational efficiency.

3. Well-structured with clear explanations of key concepts via formulas and examples.

**Weaknesses:**

1. Some parts of the method description are still unclear and require further clarification:
(1) The role of the virtual entry node needs further explanation. The paper states that a virtual entry node is created for computational convenience but does not specify details—for instance, whether all tasks can start dependency searches from this node without separately identifying tasks without predecessors. Readers may wonder what specific problem this node solves.
(2) The distinction between one-shot scheduling and step-by-step scheduling could be further compared. The paper mentions that the new method is one-shot, while traditional methods are step-by-step, but it does not clarify whether "one-shot" means all orders are arranged instantaneously, or whether dependencies are computed first before selecting tasks sequentially. Readers might misinterpret it as all tasks starting simultaneously. It should be clearly explained that "the ordering process computes all dependencies at once, while task selection remains sequential, without the need for repeated computations."
(3) Since Solution(G) represents a topological order of the DAG, different Solution(G) sequences may correspond to the same DAG. Will this many-to-one relationship have any impact on the model's performance or learning stability?
(4) The authors appear to use the EFT-greedy rule to decode Solution(G) and assign tasks to different processors. Why not integrate this process into the model itself?
(5) In Section 3.2, it is mentioned that "the attention module is applied to the reverse DAG." This point is crucial for understanding the flow of attention and should be highlighted more prominently in the main text, along with an explanation of its necessity.

2. Some aspects of the experiments are not sufficiently explored:
(1) The comparison algorithms are relatively old. If there are new algorithms from the past two years, they should be included in the comparison. If not, please explain the basis for selecting the comparison algorithms.
(2) What are the genetic algorithm parameters in demonstration learning, and how do they impact the model?
(3) Could you supplement ablation experiments to clarify how GAA-PtrNet-SA and GAA-PtrNet-GAT perform under different DAG structures?

3. The outlook could be further focused and enhanced:
Directions such as multi-objective optimization and dynamic resource adaptation lack specific technical pathways.

**Questions:**

1. The experimental section could be supplemented with descriptions of the selected datasets. The experiments used datasets such as TPC-H and Pegasus, but there is no explanation of the practical scenarios these datasets correspond to. For example, TPC-H is similar to an e-commerce data processing scenario, while Pegasus is for scientific computing. Readers who are unaware of the significance of these datasets may not understand the practicality of the experimental results.

2. Although it is mentioned that "the advantage becomes more obvious as the number of nodes increases," adding a time curve graph showing scalability as the number of nodes grows would provide a more intuitive demonstration.

3. It is recommended to add a paragraph after Figure 3 explaining why the attention scores need to be summed column-wise to serve as probabilities and justifying the rationality of this aggregation method.

4. Regarding the comparison with Jeon et al. (2023), in addition to mentioning its training instability, could further analysis be provided on the fundamental differences in model structure or objective function compared to the method proposed in this paper?

5. Include a brief outlook on how this research could be applied in the future. The paper concludes by mentioning expansion to more high-performance computing applications, but readers may not have a clear idea of specific use cases. This would help readers understand that the research is not merely theoretical but has broader practical value.

6. Figure presentation: Figure 1 illustrates the structure of the existing PtrNet and the proposed GAA-PtrNet, including their differences. My suggestion is to highlight the faster computation speed of GAA-PtrNet to emphasize the novelty and contribution of this paper. One feasible approach is to visually indicate within the dashed box that the existing PtrNet exhibits slower computation, while GAA-PtrNet performs the pair-wise node attention and aggregation operations much faster. In addition, Figure 1 could be further refined for better visual presentation.

7. Formatting and typographical issues:
(1) On the end of Page 6 and the beginning of Page 7, "Instead of using the a single makespan value" should be changed to "Instead of using the single makespan value."
(2) Multiple references appear within a single pair of parentheses in Line 48; similar issues occur elsewhere in the paper.
(3) When referring to equations, the first letter should be capitalized, e.g., Equation 1 instead of equation 1; similar issues occur elsewhere in the paper.
(4) Replace "Table 5-8" with "Figure 5-8" in Line 921.

---

> ### Author Response · Authors · 2025-11-27
> **Response to Reviewer vDpe**
>
> Thank you for your thoughtful evaluation of our paper and the valuable feedback. We are glad that the contribution of our work were recognized. In this rebuttal, we respond to your concerns and clarify several aspects of our work. We have also uploaded a revised version of our paper, in which the revised text is marked in blue.
>
> Specifically:
>
> ## Weakness 1:
> > 1.	Some parts of the method description are still unclear and require further clarification:
>
> > (1) The role of the virtual entry node needs further explanation. The paper states that a virtual entry node is created for computational convenience but does not specify details—for instance, whether all tasks can start dependency searches from this node without separately identifying tasks without predecessors. Readers may wonder what specific problem this node solves.
>
> The dummy entry node is introduced to ensure that the attention aggregation in each scheduling step can always operate on a non-empty set of already scheduled nodes. Without it, the first aggregation step would face an empty input set. At time step 0, when no nodes have been scheduled yet, we skip the aggregation and the directly select this dummy entrance node. By adding this virtual node, the GNN can also obtain meaningful information to initialize the scheduling process.
>
> We have clarified this at Line 295, Section 3.2 in the revised paper.
>
> >(2) The distinction between one-shot scheduling and step-by-step scheduling could be further compared. The paper mentions that the new method is one-shot, while traditional methods are step-by-step, but it does not clarify whether "one-shot" means all orders are arranged instantaneously, or whether dependencies are computed first before selecting tasks sequentially. Readers might misinterpret it as all tasks starting simultaneously. It should be clearly explained that "the ordering process computes all dependencies at once, while task selection remains sequential, without the need for repeated computations."
>
> We thank the reviewer for pointing out this issue. We have added further explanation at Line 94, Introduction of the revised paper to eliminate the possible misunderstandings of readers:
>
> *In general, the attention computation process obtains all priorities in one-shot, while task selection remains sequential, without the need for repeated computations of encoding and decoding of PtrNet.*
>
> >(3) Since Solution(G) represents a topological order of the DAG, different Solution(G) sequences may correspond to the same DAG. Will this many-to-one relationship have any impact on the model's performance or learning stability?
>
> Although different $Solution(G)$ sequences may correspond to the same DAG, one $Solution(G)$ only correspond to one valid execution order under the condition of fixed environment. So, we believe that there is no impact on the model's performance or learning stability.
>
> >(4) The authors appear to use the EFT-greedy rule to decode Solution(G) and assign tasks to different processors. Why not integrate this process into the model itself?
>
> This paper aims to propose a DAG scheduling method that is applicable in both single-processor and multi-processor environments. In a single-processor environment, such as TPC-H, there is no need for processor allocation, so the EFT-greedy rule is not required. In a multi-processor environment, we use the EFT-greedy rule for processor allocation. To facilitate testing in different environments, we did not integrate the EFT-greedy rule into the model.
>
> >(5) In Section 3.2, it is mentioned that "the attention module is applied to the reverse DAG." This point is crucial for understanding the flow of attention and should be highlighted more prominently in the main text, along with an explanation of its necessity.
>
> The attention module is applied to the reverse graph to ensure that the keys and queries of the attention scores conform to the causal relationship in the scheduling process and are consistent with the logic of PtrNet. In standard graph attention mechanism in message passing GNNs (e.g., GAT), a node aggregates information from its predecessors—treating the target as the query and its predecessors as keys. In PtrNet, however, the attention should represent the influence of already-scheduled nodes on candidate nodes (i.e., scheduled → candidate), which is the reverse direction of the edges. Thus, our method applies attention on the reversed DAG to preserve consistency with graph attention formulations while ensuring the computed priorities reflect the correct causal flow.
>
> We agree with the reviewer that an explanation in the main text is necessary. We have accordingly made an explicit explanation at Line 247, Section 3.2 of the revised paper.

---

> ### Author Response · Authors · 2025-11-27
>
> ## Weakness 2
>
> >Some aspects of the experiments are not sufficiently explored:
>
> >The comparison algorithms are relatively old. If there are new algorithms from the past two years, they should be included in the comparison. If not, please explain the basis for selecting the comparison algorithms.
>
> We have added the comparison experimental results with recently published SOTA one-shot DAG scheduling research [1] in Section 4 and Appendix F.
>
> [1] Qi et al., *Reinforcement learning for one-shot DAG scheduling with comparability identification and dense reward*, NeurIPS 2025.
>
> >What are the genetic algorithm parameters in demonstration learning, and how do they impact the model?
>
> We set the genetic algorithm parameters as: the size of population = problem size * 2, crossover rate = 0.15, mutation rate=0.30, generation number = 200. In the revised paper, we have reported the parameters in the “implementation details” section in Appendix E.2.
>
> The parameters do not affect the final performance of the model’s optimization; it only speeds up the convergence process. We only use the genetic algorithm to generate several near-optimal solutions for demonstration purposes. Therefore, we do not perform large-population or long-iteration evolutionary runs. Our goal is not to fully optimize with the genetic algorithm itself, but simply to obtain a small set of reasonably good solutions that can guide the early-stage exploration of the one-shot model.
>
>
> >Could you supplement ablation experiments to clarify how GAA-PtrNet-SA and GAA-PtrNet-GAT perform under different DAG structures?
>
> As shown in Fig.1(b) in the paper, our method relies on the joint work of graph attention and GAA. We first compute a global attention score matrix using the graph attention module, and then obtain the decision at each step (i.e., the pointer) through GAA. Without graph attention, we would still need a mechanism to compute attention for the candidate decisions at each step, which degenerates back to the LSTM- or context-embedding–based approaches in Fig.1(a), such as PtrNet-LSTM and PtrNet-CE that were evaluated in the experiments. Therefore, we are unable to provide additional ablation for GAA-PtrNet-SA or GAA-PtrNet-GAT.
>
> ## Weakness 3
>
> >The outlook could be further focused and enhanced:
> >Directions such as multi-objective optimization and dynamic resource adaptation lack specific technical pathways
>
> Please check our response to Question 5.
>
> ## Question 1
> >The experimental section could be supplemented with descriptions of the selected datasets. The experiments used datasets such as TPC-H and Pegasus, but there is no explanation of the practical scenarios these datasets correspond to. For example, TPC-H is similar to an e-commerce data processing scenario, while Pegasus is for scientific computing. Readers who are unaware of the significance of these datasets may not understand the practicality of the experimental results.
>
> We thank the reviewer for point out that the description of the selected datasets should be supplemented. We have revised the 2nd paragraph of Section 4.1 to include more detailed explanations about the detasets:
>
> *"TPC-H is collected from E-business oriented database query scenarios. It represents the typical structure of the computing task workflow in the E-business decision-making system. The DAG workflows in the problem instances are small but numerous."*
>
> and
>
> *"Pegasus workflows are collected from multiple scientific computing applications, including SIPHT, LIGO, GENOME, etc. Each type of application corresponds to a different workflow structure. The DAG workflows in the problem instances are large and complex."*
>
> ## Question 2
> >Although it is mentioned that "the advantage becomes more obvious as the number of nodes increases," adding a time curve graph showing scalability as the number of nodes grows would provide a more intuitive demonstration.
>
> We thank the reviewer's suggestion to better demonstrate the result with a curve graph. In the revised paper, we have added a new Figure 4 to present a more intuitive demonstration of runtime comparison.
>
> ## Question 3
> >It is recommended to add a paragraph after Figure 3 explaining why the attention scores need to be summed column-wise to serve as probabilities and justifying the rationality of this aggregation method.
>
> We have made this clarification at Line 305 in the revised paper:
>
> *The column-wise summation is used to convert the attention matrix into a valid probability distribution over candidate nodes, consistent with the pointer mechanism in PtrNet-based schedulers. Each attention score $a^{(t)}_{<i,j>}$ in the softmax-normalized matrix can be interpreted as the probability $p(<i,j>)^{(t)}$ to select the node pair $<i,j>$ from all the node pairs between $S_F^{(t)}$ and $S_C^{(t)}$, where $i∈S_F^{(t)}$ and $j∈S_C^{(t)}$. These pairwise choices are mutually exclusive, so the probability of selecting candidate node $j$ is the probability of Equation (8).*

---

> ### Author Response · Authors · 2025-11-27
>
> ## Question 4:
> >Regarding the comparison with Jeon et al. (2023), in addition to mentioning its training instability, could further analysis be provided on the fundamental differences in model structure or objective function compared to the method proposed in this paper?
>
> The method of Jeon et al. (2023) follows a typical paradigm that generates a global logits list for all task nodes by the policy network. The logits are treated as the task priorities to derive a task execution ordering via list-scheduling heuristics. However, the method of achieving sorting by generating a global list of logits inherently has the drawback of high policy gradient variance. Additionally, such a list-scheduling on global logits list suffer from the fact that multiple distinct permutations of the logits list could correspond to the same valid schedule. This many-to-one mapping biases the probability and leads to higher variance in policy gradient estimation in the learning process, which makes the training less stable and more prone to local optima. Jeon et al. (2023) further proposed to introduce Gumbel-k trick to perturb the global logits list, so that the sampled ranking better aligns with the underlying probability distribution. But this operation does not resolve the inherent many-to-one issue of list scheduling.
>
> In contrast, by achieving a one-shot PtrNet, our GAA-PtrNet omits the ranking on logits lists, and fundamentally addresses the above limitations.
>
> We have included the analysis in the “Related Work” section in the revised manuscript to highlight the differences.
>
>
> ## Question 5:
> >Include a brief outlook on how this research could be applied in the future. The paper concludes by mentioning expansion to more high-performance computing applications, but readers may not have a clear idea of specific use cases. This would help readers understand that the research is not merely theoretical but has broader practical value.
>
> Due to our method's low inference complexity, the proposed approach is suitable for latency-sensitive scheduling scenarios in cloud and cluster environments, including the scientific workflow settings represented in our benchmarks. In some application scenarios, workflow scheduling needs to consider additional QoS objectives such as cost, security, or energy efficiency. These could be naturally handled by extending our problem formulation and combining GAA-PtrNet with multi-objective reinforcement learning, which is a promising direction.
>
> Furthermore, many real-world systems require real-time scheduling of DAGs on heterogeneous resources. We have found that this field has still not been fully studied, especially in terms of its response time analysis methods and its combined application with machine learning. We believe our framework provides a potential foundation toward addressing such real-time heterogeneous scheduling problems.
>
> We have supplemented these points to better highlight the practical relevance and future potential of this research in Section “Conclusion”.
>
> ## Question 6:
> >Figure presentation: Figure 1 illustrates the structure of the existing PtrNet and the proposed GAA-PtrNet, including their differences. My suggestion is to highlight the faster computation speed of GAA-PtrNet to emphasize the novelty and contribution of this paper. One feasible approach is to visually indicate within the dashed box that the existing PtrNet exhibits slower computation, while GAA-PtrNet performs the pair-wise node attention and aggregation operations much faster. In addition, Figure 1 could be further refined for better visual presentation.
>
>  Thank you for your advice for improving the figure presentation of our paper. We have accordingly revised Figure 1 to highlight our method “pair-wise node attention generation in one-shot and attention aggregation” is fast, in order to emphasize the contribution of our work.
>
> ## Question 7:
> >Formatting and typographical issues:
> >(1) On the end of Page 6 and the beginning of Page 7, "Instead of using the a single makespan value" should be changed to "Instead of using the single makespan value."
> >(2) Multiple references appear within a single pair of parentheses in Line 48; similar issues occur elsewhere in the paper.
> >(3) When referring to equations, the first letter should be capitalized, e.g., Equation 1 instead of equation 1; similar issues occur elsewhere in the paper.
> >(4) Replace "Table 5-8" with "Figure 5-8" in Line 921.
>
> We sincerely appreciate the reviewer for carefully picking up those points. We have revised them in our manuscript, and we have thoroughly checked the whole paper for any other formatting errors and typos.

---

### Official Review · Reviewer_ppbH · 2025-10-30

**Soundness:** 3
**Presentation:** 3
**Contribution:** 3
**Rating:** 6
**Confidence:** 3

**Summary:**

This paper introduces GAA-PtrNet, a method within the Pointer Network (PtrNet) family, for optimizing Directed Acyclic Graph (DAG) workflow scheduling. The technique is specifically designed to overcome the high computational complexity and reliance on local information found in traditional PtrNet schedulers, which must repeatedly compute decoder states. The authors demonstrate that by using a graph attention aggregation (GAA) mechanism, the network can compute all pair-wise node attention scores in a single forward pass. These pre-computed scores are then sequentially aggregated during the step-by-step scheduling process to determine task probabilities, enabling solutions that are significantly faster (about 10 times) and of higher quality than previous PtrNet-based approaches.

**Strengths:**

1. Novel Scheduling Method: The paper's core novelty is its method for bypassing the sequential bottlenecks of traditional PtrNets, computing all pair-wise graph attention scores in a single forward pass ("one-shot"). This pre-computation of scores, followed by a lightweight sequential aggregation, avoids the repeated decoder or context embedding calculations required by traditional PtrNet models.

2. Strong Empirical Performance: The proposed method demonstrates superior performance in schedule quality across multiple benchmarks (TPC-H, Pegasus, and random DAGs). In experimental results, GAA-PtrNet consistently achieves a lower makespan and better "gap/%" than both previous PtrNet-based methods (PtrNet-LSTM, PtrNet-CE) and other modern learning-based schedulers.

3. Superior Scalability and Runtime: The model is empirically shown to be significantly faster, running about 10 times faster than previous PtrNet methods. This is supported by a time complexity analysis, which shows that unlike competitors, the iterative part of the scheduling algorithm is independent of the network's embedding dimension $dim$, making it scale much better as the number of nodes ($|V|$) increases.

**Weaknesses:**

1. Misleading Title and "One-Shot" Claim: While the attention score computation is indeed one-shot, the overall scheduling process remains sequential, as shown in Algorithm 1. The title and abstract could better distinguish between one-shot attention computation and stepwise decision sampling to avoid confusion.

2. Ambiguous and Inconsistent Notation ($X$ vs. $h_i$): The paper uses notation inconsistently, particularly for node features. * In Section 2.1, $X=\{x_{1},x_{2},...,x_{|V|}\}$ is defined as the raw "attribution of each task node". * In Section 2.3, $h_{i}$ is introduced without relation to $x_i$ as the "feature vectors of each task node" that are *inputs* to an LSTM encoder in prior work. * However, in Section 3.2, $H=[h_{1},h_{2},...,h_{|V|}]$ is defined as the *output* embeddings from the paper's own GNN encoder. * This reuse of $h_i$ to mean both "input features" and "output embeddings" in different sections is confusing.

3. Undefined Critical Notation ($S_F^{(t)}$): The paper uses the critical notation $S_F^{(t)}$ extensively in its core method description and key equations without providing a clear, formal definition. The reader must infer that $S_F^{(t)}$ represents the set of already scheduled nodes, primarily from a label in Figure 3 ("Mask of scheduled task nodes mask $S_{F}^{(t)}$"), which is an oversight for such a central variable.

4. Clarity of Reward Function Derivation: The paper's explanation of its training strategy in Section 3.4 could be clearer. It presents a "return-like" signal $R_{(t)}$ in Equation 9 and then presents the final "advantage" function $A_{(t)}$ in Equation 10 without explicitly detailing the derivation. The paper does not formally define the baseline equivalent of $R_{(t)}$ (e.g., $R_{(t)}^{baseline} = C(Solution(G)) - C^{heur}(v_{\pi(t)})$). By not showing how the standard advantage formula $A_{(t)} = R_{(t)} - R_{(t)}^{baseline}$ simplifies Equation 9 into Equation 10, the text requires the reader to infer these intermediate steps.

5. Deferred Citations for Core Methodology: The paper consistently defers critical citations for its core components to the appendices, harming clarity and traceability. * In Section 3.2, a "bi-directional GNN" is introduced with the generic equation $H=GNN(G)$ without an in-line citation. The specific model (Graphormer) is only identified in Appendix E.2. * Similarly, in Section 3.4, a "genetic algorithm" is named as a key part of the training strategy, but its citation is deferred to Appendix G. Providing these references in-line, rather than only in the appendix, is standard practice and would significantly improve the logical flow.

6. Missing Evaluation Sampling Details: The paper's scheduling algorithm is stochastic, relying on sampling to select nodes at each step. While the results tables report aggregated metrics like "average makespan" and "speedup", the authors never specify the number of sampling runs performed per DAG instance to compute these averages. This omission of a key experimental parameter means the statistical stability of the results is unknown, and consequently, the reported "speedup" values cannot be meaningfully interpreted.

**Questions:**

1. Please see the "Weaknesses" section for a detailed list of concerns regarding the paper's presentation, reproducibility, and theoretical clarity.

2. The recommendation in Section 3.2 to normalize attention scores across queries (before Eq. 7) is sensible but only qualitatively discussed. Could the authors clarify whether this normalization was actually implemented and whether ablations were conducted to assess its impact?

3. Have the authors tested how performance degrades when the DAG size increases substantially (e.g., > 1000 nodes)? Given the pairwise attention matrix scales quadratically, it would be helpful to quantify when this becomes computationally limiting.

---

> ### Author Response · Authors · 2025-11-27
> **Response to Reviwer ppbH**
>
> We’d like to thank for the reviewer’s careful reading, valuable suggestions and acknowledgement to our work. According to the reviewer's suggestion, we have uploaded a revised version of our paper, in which the revised text is marked in blue. Our response and rebuttals are as follows:
>
> ## Weakness 1:
> >Misleading Title and "One-Shot" Claim: While the attention score computation is indeed one-shot, the overall scheduling process remains sequential, as shown in Algorithm 1. The title and abstract could better distinguish between one-shot attention computation and stepwise decision sampling to avoid confusion.
>
> In this article, we follow the same naming convention used in previous works [1,2] and name our method as “One-Shot DAG Scheduling”. We appreciate the reviewer’s suggestion. At Line 24, abstract in the revised version, we have clarified the distinction more explicitly. Besides, to highlight the distinction, at Line 294, we have stated that:
>
> *In general, the attention computation process obtains all priorities in one-shot, while task selection remains sequential, without the need for repeated computations of encoding and decoding of PtrNet.*
>
> ***
>
> [1] Jeon et al., *Neural DAG Scheduling via One-Shot Priority Sampling*, ICLR 2023.
> [2] Qi et al., *Reinforcement learning for one-shot DAG scheduling with comparability identification and dense reward*, NeurIPS 2025.
>
> ## Weakness 2:
> >Ambiguous and inconsistent Notation ($X$ vs. $h_i$): The paper uses notation inconsistently, particularly for node features. * In Section 2.1, $X=x_1,x_2,...,x_{|V|}$ is defined as the raw "attribution of each task node". * In Section 2.3, hi is introduced without relation to xi as the "feature vectors of each task node" that are inputs to an LSTM encoder in prior work. * However, in Section 3.2, $H=[h_1,h_2,...,h_{|V|}]$ is defined as the output embeddings from the paper's own GNN encoder. * This reuse of hi to mean both "input features" and "output embeddings" in different sections is confusing.
>
> We thank the reviewer for pointing out our notation problem. To ensure consistency in notation, we have unified the transformation pipeline as
> $$x_i →  feature\ extractor + encoder\ of\ PtrNet → h_i → decoder\ of\ PtrNet$$
> to align the overall method description in Section 3 with the Preliminary (Section 2). Accordingly, we have revised Section 2.3 and Equation (1) to clarify that the LSTM encoder in prior work takes $X=\{x_1,x_2,…,\} $ as its input, rather than $h_i$.
>
> ## Weakness 3:
>
> >Undefined Critical Notation ($S_F^{(t)}$): The paper uses the critical notation $S_F^{(t)}$ extensively in its core method description and key equations without providing a clear, formal definition. The reader must infer that $S_F^{(t)}$ represents the set of already scheduled nodes, primarily from a label in Figure 3 ("Mask of scheduled task nodes mask $S_F^{(t)}$"), which is an oversight for such a central variable.
>
> We thank the reviewer for pointing out this omission. $S_F^{(t)}$ denotes the set of tasks that have already been scheduled at time step $t$. We acknowledge that it should have been more explicitly defined in the main text. We have added a definition statement at Line 202, Section 3.1 to improve clarity.
>
> ## Weakness 4:
>
> >Clarity of Reward Function Derivation: The paper's explanation of its training strategy in Section 3.4 could be clearer. It presents a "return-like" signal $R(t)$ in Equation 9 and then presents the final "advantage" function $A(t)$ in Equation 10 without explicitly detailing the derivation. The paper does not formally define the baseline equivalent of $R(t)$ (e.g., $R(t)baseline=C(Solution(G))−C^{heur}(v_{π(t)}))$. By not showing how the standard advantage formula $A(t)=R(t)−R(t) baseline$ simplifies Equation 9 into Equation 10, the text requires the reader to infer these intermediate steps.
>
> We thank the reviewer for pointing out this. Baseline is the form of $R(t)$ obtained under the heuristic algorithm, which is:
> $$baseline=C(Solution(G))−C^{heur}(v_{π(t)}))$$
> We have added this equation in the revised manuscript. Due to page limit in the revised version, we have simplified this section and moved the equations to Appendix G.

---

> ### Author Response · Authors · 2025-11-27
>
> ## Weakness 5:
>
> >Deferred Citations for Core Methodology: The paper consistently defers critical citations for its core components to the appendices, harming clarity and traceability. * In Section 3.2, a "bi-directional GNN" is introduced with the generic equation $H=GNN(G)$ without an in-line citation. The specific model (Graphormer) is only identified in Appendix E.2. * Similarly, in Section 3.4, a "genetic algorithm" is named as a key part of the training strategy, but its citation is deferred to Appendix G. Providing these references in-line, rather than only in the appendix, is standard practice and would significantly improve the logical flow.
>
> We thank the reviewer for pointing out this issue. We have added the deferred citations in the main text in the revised manuscript according to the reviewer’s suggestion.
>
> ## Weakness 6:
> >Missing Evaluation Sampling Details: The paper's scheduling algorithm is stochastic, relying on sampling to select nodes at each step. While the results tables report aggregated metrics like "average makespan" and "speedup", the authors never specify the number of sampling runs performed per DAG instance to compute these averages. This omission of a key experimental parameter means the statistical stability of the results is unknown, and consequently, the reported "speedup" values cannot be meaningfully interpreted.
>
> We train the model in parallel on 16 sampled problem instances as one batch, with up to 5000 episodes per batch. We have added these detailed descriptions at the end of Section 4.1.
>
> ## Question 1:
> >Please see the "Weaknesses" section for a detailed list of concerns regarding the paper's presentation, reproducibility, and theoretical clarity.
>
> Please check our response to the weaknesses.
>
> ## Question 2:
> > The recommendation in Section 3.2 to normalize attention scores across queries (before Eq. 7) is sensible but only qualitatively discussed. Could the authors clarify whether this normalization was actually implemented and whether ablations were conducted to assess its impact?
>
>
> In Appendix F.4 in the revised paper, we reported the ablation about the column-wise normalization on the pair-wise node attentions. The results show that the normalization of the attention scores can lead to faster converge.
>
> ## Question 3:
> > Have the authors tested how performance degrades when the DAG size increases substantially (e.g., > 1000 nodes)? Given the pairwise attention matrix scales quadratically, it would be helpful to quantify when this becomes computationally limiting.
>
> We have tested the performance for the case that the node number are more than 1000. We have reported the results in Table 3 of the article. In the TPC-H workflow, the average number of nodes for each DAG is 9.17, and the number of nodes under TPC-H 150 configuration (150 sub-DAGs) is greater than 1000.

---

### Official Review · Reviewer_zot3 · 2025-11-01

**Soundness:** 2
**Presentation:** 3
**Contribution:** 1
**Rating:** 2
**Confidence:** 4

**Summary:**

This paper addresses the *one-shot DAG scheduling* problem, where a neural network generates a complete task execution order in a single forward pass by assigning priority scores to all nodes. The proposed method, **GAA-PtrNet**, replaces the LSTM-based encoder in standard Pointer Networks with a GNN-based encoder and adopts a learning framework for priority-based DAG scheduling. The authors claim two contributions: (1) the novel network architecture enabling one-shot PtrNet-based scheduling, and (2) a training strategy using policy gradient RL with dense rewards and demonstration learning.

However, these contributions are incremental. The GNN encoder has already been explored in prior work, such as [1] Jeon et al. (ICLR 2023) and [2] Qi et al. (NeurIPS 2025). Furthermore, the training framework closely follows [2]. The experimental improvements reported in Tables 5–6 are marginal, and the overall setup lacks comprehensiveness.

----
[1] Jeon et al., *Neural DAG Scheduling via One-Shot Priority Sampling*, ICLR 2023.

[2] Qi et al., *Reinforcement learning for one-shot DAG scheduling with comparability identification and dense reward,* NeurIPS2025.

**Strengths:**

* The paper applies the Pointer Network framework to DAG scheduling with a graph-aware encoder, providing a unified neural approach to one-shot scheduling.
* The methodology is clearly presented.
* The use of GNNs enables the model to capture structural task dependencies more effectively than sequence-based LSTMs.

**Weaknesses:**

* **Limited novelty**: The core innovation of using GNNs for one-shot DAG scheduling has been previously explored in recent works, particularly [1] Jeon et al. (ICLR 2023) and [2] Qi et al. (NeurIPS 2025).
* **Similar training**: Several technical sections (e.g., Section 3.4) appear conceptually similar to [2] but are not properly cited, raising concerns about originality.
* **Overclaimed contributions**: The claim of being “the first PtrNet-based one-shot DAG scheduler” is overstated and contradicts the authors’ own acknowledgement of prior PtrNet-based DAG scheduling studies. One-shot scheduling with graph attention-enhanced has been previously demonstrated in [1] and [2].
* **Insufficient experimental validation**:
  - Performance improvements on larger-scale problems (e.g., LIGO-400 in Table 5) are minimal (<1%), suggesting limited scalability advantage.
  - The comparison with [2] is notably absent from the results tables, despite using similar benchmarks and baselines. Independent comparison shows that GAA-PtrNet does not consistently outperform [2] across all metrics (e.g., TPC-H 50, 100; SIPHT-200, 300).
  - Limited benchmark diversity: Evaluation is missing on several important Pegasus workflows (Montage, CyberShake, Inspiral) and larger-scale instances (e.g., SIPHT-1000).
  - Discrepancies in reported results (e.g., different gap calculations for the same SIPHT-400 makespan values between this paper and [2]) raise concerns about experimental rigor.
* **Inappropriate baseline selection**: The use of POMO-DAG, originally designed for routing problems, rather than comparing against more recent and relevant learning-based scheduling solvers from top-tier conferences/journals, limits the persuasiveness of the performance claims.

**Questions:**

1. Section 3.4 appears highly similar to Section 4.3 of Qi et al. (NeurIPS 2025), yet [2] is not cited. Could the authors explain the methodological differences and justify how this section constitutes a contribution?
1. Why are the baselines chosen POMO (which was designed for routing problems)? Please compare against recent neural DAG or workflow schedulers in top-tier conferences or journals, reporting both performance and inference time.
1. The reported improvements in LIGO (Table 5) are below 2%, and gains diminish as node size increases. Could the authors analyze why the method scales poorly and whether the network saturates for larger DAGs?
1. Could the method be evaluated on additional Pegasus datasets (e.g., Montage, CyberShake, Inspiral) and job shop scheduling benchmarks to test robustness and generality?
1. Since the datasets and settings are nearly identical to [2], please include [2]’s reported results in your tables for a direct comparison.  Also, why were the results on SIPHT-1000 omitted?

---

> ### Author Response · Authors · 2025-11-27
> **Resonse to Reviewer zot3**
>
> Considering the reviewer's concerns, we have uploaded a revised version of our paper, in which the revised text is marked in blue. Specifically, our response and rebuttals are as follows:
>
> ## Weakness 1:
>
> > Limited novelty: The core innovation of using GNNs for one-shot DAG scheduling has been previously explored in recent works, particularly [1] Jeon et al. (ICLR 2023) and [2] Qi et al. (NeurIPS 2025).
>
> Although [1] and [2] have explored the use of GNNs for one-shot DAG scheduling, some important issues remain unsolved.
>
> 1. One-shot GNN+RL methods typically rely on generating a global logits list of logits for all task nodes by the policy network. The logits are treated as the task priorities to derive a task execution ordering via list-scheduling heuristics. However, **the scheduling method which achieves ranking by generating a global list of logits inherently has large policy gradient variance**. Additionally, such list-scheduling based on global logits list suffers from the fact that multiple distinct permutations of the logits list may correspond to the same valid schedule. **This many-to-one mapping biases the probability of policy sampling, so that policy gradient estimation is biased in the learning process, which makes the scheduler training more prone to local optima**. [1] introduces the Gumbel top-k trick to perturb the global logits list, so that the sampled ranking better aligns with the underlying probability distribution. But this mechanism does not resolve the inherent many-to-one issue of list scheduling. [2] proposed a comparable antichain identification mechanism, from the perspective of reducing redundant pairwise comparisons among logits during ranking. **These two methods only partly address this issue.** Their schedulers still depend on generating a global logits list to rank the task nodes.
> 2. Some studies achieved DAG scheduling through PtrNet. It does not rely on producing a global logits list for ranking, and therefore avoids the bias of probability of policy sampling in ranking-based one-shot methods. However, these PtrNet-based scheduling methods need to repeatedly compute the decoder's hidden state or context embeddings according to the recent local decisions, which leads to **limited capability of exploiting the DAG global topological structure, high computation complexity and inability to achieve one-shot inference in scheduling**.
> Different from previous studies, **our GAA-PtrNet fundamentally addresses the above challenges from the perspective of omitting the ranking on logits lists and achieving a one-shot PtrNet-based scheduler**. Specifically, GAA-PtrNet innovatively aggregates the graph attention scores among task nodes after extracting the task node features and computing the graph attention in one-shot, which are directly used for calculating the policy sampling probabilities, thereby realize the one-shot PtrNet DAG scheduler.
>
> We have reported comparison experiment results with [2] (including the experiments on the configuration of 1000-sized situations). The results are listed in the below Table W.1.1 (optimization performance) and Table W.1.2 (runtime). We have also added these results in Table 2,3,4 and 5,6 (Appendix F) in the revised paper.The supplemented results show that our method achieves superior optimization performance in most cases, especially in structurally complex workflows (Tables 5–6 in Appendix F.1 of the revised paper), highlighting its stronger ability to handle with scheduling in complex high-performance computation scenarios. Moreover, our method exhibits lower runtime across settings (Figure 4 and Table 4), providing advantages in scheduling for latency-sensitive applications. We believe that the additional analyses, clarifications, and new experiments substantiate that our study is not an incremental contribution, but represents a meaningful and breakthrough contribution in the one-shot DAG scheduling domain
>
> We have also accordingly added descriptions in the Introduction and Related Work sections in the revised version so that the readers can clearly understand our contributions.

---

> ### Author Response · Authors · 2025-11-27
>
> Table W.1.1. Comparison experiments results in optimization performanc including [2], ms is the abbreviation of makespan.:
>
> |Method|SIPHT-100|||SIPHT-200|||SIPHT-300|||SIPHT-400|||SIPHT-1000|||
> |---|---|---|---|---|---|---|---|---|---|---|---|---|---|---|---|
> | |ms| gap| speed up|ms|gap|speed up| ms| gap|speed up|ms|gap|speed up|ms|gap|speed up|
> |GAA-PtrNet-SA|**191.1**| **-15.81**|**2.43**| 340.3| -4.89| 2.45| 542.1| -0.20| 2.51| 708.5| -0.88| 2.51| 1818.8| -0.14| 2.51|
> |GAA-PtrNet-GAT|191.7|-15.55|2.42|338.8|-3.41| 2.51|542.7|-0.05|2.50|**708.3**|**-0.91**| **2.51**| **1818.6**|**-0.15**|**2.51**|
> |[1]|218.5| -3.74| 2.23| 352.2| -1.57| 2.42| 550.6| 1.40| 2.47| 712.7| -0.29| 2.50| 1898.1| 4.21| 2.40|
> |[2]|196.9| -13.26| 2.36| **338.4**|**-5.42**| **2.51**| **541.6**|**-0.25**| **2.51**| **708.3**| **-0.91**| **2.51**| 1819.2| -0.13| 2.51|
> |**Method**|  **LIGO-100** | ||**LIGO-200**  |||**LIGO-300**|||**LIGO-400**|||**LIGO-1000**|||
> | | ms| gap | speed up| ms| gap | speed up| ms | gap |speed up | ms | gap |speed up | ms | gap |speed up |
> |GAA-PtrNet-SA|**211.0**| **-2.98**|**2.49**| 460.7| -0.48| 2.49| 666.9| -0.51| 2.50| 956.9| -0.25| 2.50| 2374.7| 0.05| 2.50|
> |GAA-PtrNet-GAT|211.8| -2.62| 2.48| **460.6**| **-0.50**| **2.50**| **666.1**| **-0.63**| **2.51**| **956.6**| **-0.28**| **2.50**| 2374.2| 0.03| 2.50|
> |[1]|217.0| -0.23| 2.42| 465.1| 0.47| 2.48| 670.8| 0.07| 2.49| 962.8| 0.36| 2.49| 2376.7| 0.13| 2.50|
> |[2]|214.0| -1.61| 2.46| 462.1| -0.17| 2.50| 668.8| -0.22| 2.49| **956.6**| **-0.28**| **2.50**| **2373.5**| 0| 2.50|
> |**Method**| **GENOME-100**|||**GENOME-200**  |||**GENOME-300**|||**GENOME-400**|||**GENOME-1000**|||
> | | ms| gap | speed up| ms| gap | speed up| ms | gap |speed up | ms | gap |speed up | ms | gap |speed up |
> |GAA-PtrNet-SA|**2435.5**| **-4.61**| **2.44**| 2351.2|-0.94|2.46|4723.3|-0.60|2.48|**3451.1**|**-0.06**|**2.48**|**14948.1**| **-0.46**|**2.50**|
> |GAA-PtrNet-GAT|2438.4|-4.49|2.44|**2348.1**|**-1.07**|**2.47**|**4721.4**|**-0.64**|**2.48**|3473.5|0.559|2.47|14966.2|-0.34| 2.49|
> |[1]|2511.4|-1.63| 2.37| 2369.9| -0.15| 2.48| 4755.3| 0.07| 2.46| 3483.2| 0.87| 2.46| 15001.9| -0.10| 2.49|
> |[2]|2468.2| -3.32| 2.41| 2350.5| -0.96| 2.47| 4728.4| -0.49| 2.48| 3453.0| -0.01| 2.48| 14955.8| -0.41| 2.50|
> | **Method**|  **TPC-H 50** | ||     **TPC-H 100**  |||   **TPC-H 150**   |||  |||     |||
> | | ms| gap | speed up| ms| gap | speed up| ms | gap |speed up | |  | | |  | |
> |GAA-PtrNet-SA|21.37| -14.42| 5.25| 39.59| -7.54| 5.37| **67.08**| **-3.84**| **4.81**|||||| |
> |GAA-PtrNet-GAT|21.33| -14.58| 5.26| 42.18| -1.49| 5.04| 67.81| -2.79| 4.96|||| | ||
> |[1]|23.73| -4.95| 4.73| 41.22| -3.81|5.15|74.02|6.11|4.35| | | | | | |
> |[2]|**20.49**|**-17.73**|**5.47**|**39.22**|**-8.47**|**5.42**| 73.47| 5.32| 4.39| | | | | | |
>
> Table W.1.2. Comparison experiments results in runtime including [2]:
>
> |Method|size=100| ||size=200|||size=300|||size=400|||size=1000|||
> |---|---|---|---|---|---|---|---|---|---|---|---|---|---|---|---|
> | |SIPHT|LIGO|GENOME|SIPHT|LIGO|GENOME|SIPHT|LIGO|GENOME|SIPHT|LIGO|GENOME|SIPHT|LIGO|GENOME|
> |GAA-PtrNet-SA|0.126| 0.129| 0.130| 0.219| 0.226| 0.225| 0.292| 0.300| 0.299| 0.347| 0.356| 0.354| 0.985| 0.907| 0.963|
> |GAA-PtrNet-GAT|0.136| 0.128| 0.129| 0.228| 0.226| 0.223| 0.301| 0.297| 0.298| 0.355| 0.357| 0.356| 0.928| 0.734| 0.966|
> |[1]|0.07| 0.08| 0.08| 0.15| 0.16| 0.15| 0.26| 0.26| 0.27| 0.43| 0.43| 0.46| 2.44| 2.39| 2.58|
> |[2]|0.61| 0.62| 0.61| 0.97| 0.99| 1.00| 1.17| 1.20| 1.20| 1.22| 1.26| 1.29| 2.59| 2.59| 2.79|
>
> ## Weakness 2:
>
> > Similar training: Several technical sections (e.g., Section 3.4) appear conceptually similar to [2] but are not properly cited, raising concerns about originality.
>
> Since by the submission deadline of ICLR 2026, there is no publicly available version of [2] that can be cited. Section 3.4 part only serves as a minor contribution. According to the reviewer’s suggestion, we added the citation of [2], and simplify the description in Section 3.4 in the revised version.
>
> ## Weakness 3:
>
> > Overclaimed contributions: The claim of being “the first PtrNet-based one-shot DAG scheduler” is overstated and contradicts the authors’ own acknowledgement of prior PtrNet-based DAG scheduling studies. One-shot scheduling with graph attention-enhanced has been previously demonstrated in [1] and [2].
>
> The original description may have caused misunderstanding. We do not claim that we are the “**first one-shot DAG scheduler**”, but the “**first PtrNet-based one-shot DAG scheduler**”. Specifically, our GAA-PtrNet innovatively aggregates the graph attention scores among task nodes after extracting the task node features and computing the graph attention in one-shot, which are directly used for calculating the policy sampling probabilities, thereby realize the one-shot PtrNet DAG scheduler. In addition, we would like to argue that **the methods of [1] and [2] are not based on graph-attention-enhancing**.
>
> We have accordingly revised the Introduction and Related Work section.

---

> ### Author Response · Authors · 2025-11-27
>
> ## Weakness 4:
>
> > Insufficient experimental validation:
>
> > (1) Performance improvements on larger-scale problems (e.g., LIGO-400 in Table 5) are minimal (<1%), suggesting limited scalability advantage.
>
> We’d like clarify that the task nodes in some larger workflows (e.g., LIGO-400) has less attribution diversity and makes the scheduler’s decisions less different, leading to limited room for improvement. Our method is particularly suitable for more complex tasks, as demonstrated by the additional results on 1000-sized instances and randomly generated DAG workflows controlled via adjustable parameters.  These results highlight the advantages of our approach in handling complex high-performance computing scenarios.
>
> In the meanwhile, we not only emphasize the improvement in the optimization goals, but also highlight our advantages in the runtime aspect. Our method consistently achieves lower runtime, highlighting its computational complexity advantage as a one-shot scheduling approach.
>
>
> > (2) The comparison with [2] is notably absent from the results tables, despite using similar benchmarks and baselines. Independent comparison shows that GAA-PtrNet does not consistently outperform [2] across all metrics (e.g., TPC-H 50, 100; SIPHT-200, 300).
>
> Since by the submission deadline of ICLR 2026, there is no publicly available version of [2] that can be cited. We have presented the comparison results with [2] in Table W.1.1 and Table W.1.1, and we have added these results in the revised paper.
>
> Across most settings, our method achieves better optimization performance than [2], while it may underperform in some specific instances. We believe this is because the structure and scale of these workflows result in fewer long dependency constraints, which makes the problem of policy gradient estimation caused by the logits list less prominent. Therefore, the advantage of removing the logits list is not significant, and thus our method does not demonstrate any superiority. However, our method still achieves a consistent advantage in runtime. We attribute this to the fact that, although [2] is also a one-shot scheduler, its comparable antichain identification over the global logits list introduce additional computational complexity.
>
>
> > (3) Limited benchmark diversity: Evaluation is missing on several important Pegasus workflows (Montage, CyberShake, Inspiral) and larger-scale instances (e.g., SIPHT-1000).
>
> We have added experimental results about optimization objective on larger scale problem instances (1000-sized) in Table W.4.1, and these results is added to the additional columns in Table 2 and Table 5,6 in Appendix F.1 of the revised paper. We have also reported the runtime performance on 1000-sized problem instances in Table W.1.2 that is presented above, which is added to Table 4 in the revised paper.
>
> We have also added experimental results on other Pegasus workflows in Table W.4.2, which is also added to Table 7, Appendix F.1 in the revised paper. However, due to the small variance in computational workload among task nodes in these workflows, under fundamental DAG scheduling setting, different methods produce almost indistinguishable results among different methods. For this reason, we only briefly report the makespan. In contrast, on workflows with more diverse structures.
>
> Table W.4.1. Experimental results about optimization objective on 1000-sized problem instances, ms is the abbreviation of makespan.::
>
> | Method |  SIPHT-1000 | ||     LIGO-1000  |||   GENOME-1000    |||
> |-----------------|-----------------|-----------------|-----------------|-----------------|-----------------|-----------------|-----------------|-----------------|-----------------|
> | | ms| gap     | speed up      | ms| gap | speed up      | ms | gap | speed up      |
> | GAA-PtrNet-SA    | 1818.8  | -0.14  | 2.51  | 2374.7  | 0.05       | 2.50  | **14948.1** | **-0.46** | **2.50** |
> | GAA-PtrNet-GAT   | **1818.6** | **-0.15** | **2.51** | 2374.2  | 0.03       | 2.50  | 14966.2  | -0.34  | 2.49  |
> | PtrNet-LSTM       | 1829.2  | 0.40   | 2.49 | 2375.5  | 0.08       | 2.50  | 14982.2  | -0.20  | 2.49  |
> | PtrNet-CE       | 1832.0  | 0.58   | 2.50   | 2394.7  | 0.89       | 2.48  | 15007.9  | -0.06  | 2.49  |
> | HEFT (heuristic) | 1821.4  | -      | 2.51   | 2373.5  | -  | 2.50 | 15016.8  | -      | 2.48  |
> | Jeon2023 | 1898.1  | 4.21   | 2.40  | 2376.7  | 0.13       | 2.50  | 15001.9  | -0.10  | 2.49  |
> | Qi2025   | 1819.2| -0.13  | 2.51  |  **2373.5** | **0** | **2.50**   | 14955.8  |-0.41  | 2.50  |
> | POMO-DAG | 1875.1  | 2.95   | 2.44  | 2382.0  | 0.36       | 2.50  | 15005.7  | -0.07  | 2.49  |
> | EGS      | 1821.0  | -0.02  | 2.51  | **2373.5** | **0** | **2.50** | 14970.4  | -0.31  | 2.49  |

---

> ### Author Response · Authors · 2025-11-27
>
> Table W.4.2. Experimental results of makespan on other Pegasus workflows:
>
> | Method |  MONTAGE| ||      ||CYBERSHAKE|       ||||
> |-----------------|-----------------|-----------------|-----------------|-----------------|-----------------|-----------------|-----------------|-----------------|-----------------|---|
> | | **100**|**200**|**300**|**400**|**1000**|**100**|**200**|**300**|**400**|**1000**|
> |GAA-PtrNet-SA| 13.8| 26.8| 40.3| 54.4| 135.1| 28.2| 50.1| 76.3| 100.5| 252.7|
> |GAA-PtrNet-GAT| 13.8| 27.0| 40.3| 54.4| 135.2| 28.2| 50.2| 76.4| 100.6| 253.0|
> |PtrNet-LSTM| 13.9| 26.8| 40.6| 54.4| 135.1| 28.2| 50.3| 76.5| 100.5| 252.7|
> |PtrNet-CE| 14.0| 26.8| 40.6| 54.4| 135.1| 28.3| 50.3| 76.5| 100.6| 253.0|
> |HEFT (heuristic)| 13.9| 27.0| 40.5| 54.6 |135.3| 28.6| 50.9| 77.4 |101.6| 255.0|
> |[1]|13.9| 27.0| 40.5| 54.6| 135.3| 28.6| 50.9 |77.4| 101.6| 255.0|
> |[2]|13.9| 26.9| 40.4| 54.4| 135.1| 28.3| 50.1| 76.3| 100.7| 252.9|
>
> > (4) Discrepancies in reported results (e.g., different gap calculations for the same SIPHT-400 makespan values between this paper and [2]) raise concerns about experimental rigor.
>
> In our research, we calculate the gap according to:
> $$Gap = \frac{makespan – heuristic\ makespan}{heuristic\ makespan}$$
> And thus obtained the gap “0.91” for SIPHT-400 in Table 2 in our paper. We think that the gap reported in [2] might be a typo. We have always recognized the seriousness of academic misconduct and have always been committed to ensuring the authenticity of our work.
>
>
> ## Weakness 5:
>
> > Inappropriate baseline selection: The use of POMO-DAG, originally designed for routing problems, rather than comparing against more recent and relevant learning-based scheduling solvers from top-tier conferences/journals, limits the persuasiveness of the performance claim.
>
> POMO-DAG is a classical learning-based approach for combinatorial optimization and can indeed be applied to DAG scheduling. In addition, according to the reviewer’s suggestion, we have conducted comparison experiments with the recently published [2] and reported the resuits in Section 4 and Appendix F in the revised paper.
>
> ## Question 1:
>
> >Section 3.4 appears highly similar to Section 4.3 of Qi et al. (NeurIPS 2025), yet [2] is not cited. Could the authors explain the methodological differences and justify how this section constitutes a contribution?
>
> Since by the submission deadline of ICLR 2026, there is no publicly available version of [2] that can be cited. Section 3.4 part only serves as a minor contribution. We have followed the reviewer’s suggestion to add a citation of [2], and simplify the description in Section 3.4 in the revised version.
>
> ## Question 2:
>
> > Why are the baselines chosen POMO (which was designed for routing problems)? Please compare against recent neural DAG or workflow schedulers in top-tier conferences or journals, reporting both performance and inference time.
>
> Please check our response to Weakness 5.
>
> ## Question 3:
>
> >The reported improvements in LIGO (Table 5) are below 2%, and gains diminish as node size increases. Could the authors analyze why the method scales poorly and whether the network saturates for larger DAGs?
>
> Please check our response to Weakness 4.1.

---

> ### Author Response · Authors · 2025-11-27
>
> ## Question 4:
>
> > Could the method be evaluated on additional Pegasus datasets (e.g., Montage, CyberShake, Inspiral) and job shop scheduling benchmarks to test robustness and generality?
>
> Results on additional Pegasus are in Table W.4.1 aboue. For job shop scheduling problem (JSSP), the results are in Table Q.4.1, which is also added in the Table 8, Appendix F.3 in the revised paper. The results show that our method obviously outperforms the baselines. This is because the baselines rely solely on a global priority list over operations. When applying a global priority list, many operations have to wait for higher-priority operations on the same machine, leading to much longer machine idle time.  In contrast, our method is based on PtrNet, which dynamically derives the optimal scheduling decision for each idle machine at every decision point through attention aggregation, thereby avoiding the above issue.
>
> Table Q.4.1. Experimental results of makespan on JSSP:
>
> | **Method**         |**JSSP 20\*10**|**JSSP 20\*20**|**JSSP 30\*10**|**JSSP 30\*20**|
> |-------------------------|----------|----------|----------|----------|
> | GAA-PtrNet-SA | 181.9    | 283.4    | 275.8    | 387.0    |
> | GAA-PtrNet-GAT | 194.3    | 284.7    | 269.7    | 391.8    |
> | PtrNet-LSTM    | 217.4    | 313.3    | 321.3    | 428.1    |
> | PtrNet-CE      | 217.5    | 303.4    | 286.1    | 433.0    |
> | SPT (Baseline)          | 516.7    | 1096.2   | 845.9    | 1692.0   |
> | Jeon et al.        | 445.2    | 964.0    | 735.6    | 1548.6   |
> | Qi et al.         | 397.3    | 813.8    | 571.3    | 1426.0   |
> | POMO-DAG                | 341.6    | 936.3    | 971.3    | 1458.0   |
> | EGS                     | 465.9    | 1034.5   | 837.5    | 1604.1   |
>
> ## Question 5:
>
> > Since the datasets and settings are nearly identical to [2], please include [2]’s reported results in your tables for a direct comparison. Also, why were the results on SIPHT-1000 omitted?
>
> In the revised paper, we have followed the reviewer's suggestion to cite [2], compare its results in the experiments, and conducted experiments on larger-scaled problem instances (SIPHT-1000).
>
> ***
>
> [1] Jeon et al., *Neural DAG Scheduling via One-Shot Priority Sampling*, ICLR 2023.
>
> [2] Qi et al., *Reinforcement learning for one-shot DAG scheduling with comparability identification and dense reward*, NeurIPS 2025.

---

### Official Review · Reviewer_c4UY · 2025-11-02

**Soundness:** 2
**Presentation:** 2
**Contribution:** 1
**Rating:** 2
**Confidence:** 4

**Summary:**

This paper addresses limitations in Pointer Network (PtrNet)-based methods for Directed Acyclic Graph (DAG) workflow scheduling, which suffer from an inability to exploit global DAG structure and high computational complexity due to their sequential decoding. The key technical contribution is GAA-PtrNet, a novel model that replaces the sequential decoder with a graph attention aggregation (GAA) mechanism to compute node priorities in a single, one-shot forward pass. This innovation omits the explicit decoder structure, enabling the model to capture global topological information and drastically reduce complexity. For training, a policy gradient RL strategy with dense rewards and demonstration learning is designed. Experimentally, the proposed method not only produces superior scheduling solutions but also runs approximately 10 times faster than prior PtrNet-based approaches and outperforms other learning-based methods.

**Strengths:**

1. The paper is well-organised and presented in a way that makes it quite easy for the audience (even those with limited knowledge of workflow scheduling) to follow, with a clear blueprint of the proposed method and key components, along with their functionality, in mind.

2. Detailed analysis of the computational complexity is provided.

**Weaknesses:**

1. The idea of using the graph neural network and reinforcement learning to solve scheduling problems has been widely investigated, making the contribution of this paper very limited.

2. The paper will have limited inspiration for the learning-for-scheduling domain.

**Questions:**

1. Will the method be capable of handling dynamic workflow scheduling? Or other scheduling problem variants? Like the one proposed in the paper: https://openreview.net/pdf?id=4PlbIfmX9o

---

> ### Author Response · Authors · 2025-11-27
> **Response to Reviewer c4UY**
>
> Our response and rebuttals are as follows:
>
> ## Response to the Weaknesses:
>
> >The idea of using the graph neural network and reinforcement learning to solve scheduling problems has been widely investigated, making the contribution of this paper very limited.
>
> >The paper will have limited inspiration for the learning-for-scheduling domain.
>
> Although the use of GNNs and RL for scheduling has been extensively studied, some important challenges in this domain remain unresolved. Due to page limits, we didn’t stated them clearly, which causes the reviewer’s confusion. Here, we’d like to make detailed clarification:
>
> 1. Earlier GNN+RL approaches follow a Markov decision process and generate a schedule step by step, requiring repeated extraction of global environment features and recalculation of schedules, which results in high computational complexity. Some recent work addressed this issue by developing one-shot neural schedulers, which generate all the sub-decisions through a single forward propagation of the network, thereby reducing computation cost. These one-shot GNN+RL methods typically rely on generating a global logits list for all task nodes by the policy network. The logits list is treated as the task priority list to derive a task execution ordering via list-scheduling heuristics. However, **the scheduling method which achieves ranking by generating a global list of logits inherently has large policy gradient variance**. Additionally, such list-scheduling based on global logits list suffers from the fact that multiple distinct permutations of the logits list may correspond to the same valid schedule. **This many-to-one mapping biases the probability of policy sampling, so that policy gradient estimation is biased in the learning process, which makes the scheduler training more prone to local optima. A recent work in NeurIPS 2025 [1] proposed a comparable antichain identification mechanism, from the perspective of reducing redundant pairwise comparisons among logits during ranking, partly addressed this issue. However, the scheduler still depends on generating a global logits list to rank the task nodes.**
> 2. Some studies achieved DAG scheduling through PtrNet. It does not rely on producing a global logits list for ranking, and therefore avoids the bias of probability of policy sampling in ranking-based one-shot methods. However, these PtrNet-based scheduling methods need to repeatedly compute the decoder's hidden state or context embeddings according to the recent local decisions, which leads to **limited capability of exploiting the DAG global topological structure, high computation complexity and inability to achieve one-shot scheduling**.
>
> Different from previous studies, **our GAA-PtrNet fundamentally addresses the above challenges from the perspective of omitting the ranking on logits lists and achieving a one-shot PtrNet-based scheduler**. Specifically, GAA-PtrNet innovatively aggregates the graph attention scores among task nodes after extracting the task node features and computing the graph attention in one-shot, which are directly used for calculating the policy sampling probabilities, thereby realize the one-shot PtrNet DAG scheduler.
>
> **We have also conducted additional comparison experiments with the recently published [1]**. The results are added in Table 2,3,4 and Appendix F in the revised paper. The supplemented results show that our method achieves superior optimization performance in most cases, especially in structurally complex workflows (Tables 5–6 in Appendix F.1 of the revised paper), highlighting its stronger ability to handle with scheduling in complex high-performance computation scenarios. Moreover, our method exhibits lower runtime across settings (Figure 4 and Table 4), providing advantages in scheduling for latency-sensitive applications. We believe that the proposed method provides a breakthrough contribution for learning-based DAG scheduling.
>
> **We have accordingly revised the Introduction and Related Work sections and submitted a revised version, so that the readers can clearly understand our contributions. The revised texts are marked in blue.**
>
> ***
>
> [1] Qi et al., *Reinforcement learning for one-shot DAG scheduling with comparability identification and dense reward*, NeurIPS 2025.
>
> ## Response to the Question:
>
> > Will the method be capable of handling dynamic workflow scheduling? Or other scheduling problem variants? Like the one proposed in the paper: https://openreview.net/pdf?id=4PlbIfmX9o
>
> Regarding dynamic or variant scheduling scenarios, our GAA-based design can be readily extended: Due to its low inference complexity, the proposed approach is suitable for latency-sensitive scheduling scenarios, making it promising for online scheduling settings. While such extensions are beyond the current paper’s scope, we plan to explore them in future work.

---

### Author Response · Authors · 2025-12-03
**Author Rebuttal by Authors**

Below is a brief summary of our rebuttal. We have addressed all reviewer comments, and made revisions in the newly submitted paper. The revised texts are marked in blue.

We believe that reviewers **ppbH** and **vDpe** thoroughly understood our core contribution and acknowledged the novelty of our approach.

+ Reviewer **ppbH** ’s main concerns are the clarity of article presentations, details about the experimental settings, and the necessity to add an ablation study regarding a key process “normalize attention scores across queries” in GAA. We have addressed these issues in the new version, and added the results of the corresponding ablation experiments in Appendix F.4.

+ Reviewer **vDpe** ’s main concerns are several description details about our methodology, and suggestions to improve our presentation of the figures and tables. We have updated the manuscript accordingly. In addition, reviewer **vDpe** inquired whether the ablation experiments for GAA-PtrNet-SA and GAA-PtrNet-GAT can be supplemented. However, as shown in Fig.1(b) in our paper, our method relies on the joint design of graph attention and GAA, and each component cannot work independently. We first compute a global attention score matrix using the graph attention module, and then obtain the decision at each step (i.e., the pointer) through GAA. Without graph attention, we would still need a mechanism to compute attention for the candidate decisions at each step, which degenerates back to the LSTM- or context-embedding–based approaches in Fig.1(a), such as PtrNet-LSTM and PtrNet-CE that were evaluated in the experiments. Therefore, we are unable to provide additional ablation for GAA-PtrNet-SA or GAA-PtrNet-GAT.

We think that reviewer **zot3** and **c4UY** did not fully understand our paper, and mistook our contribution as being incremental or limited.

+ Reviewer **zot3** raised the concern about the contribution and distinction of our work compared with the NeurIPS 2025 paper [2] that was published after the submission deadline of ICLR 2026, and mistook our contribution as being incremental compared to the recent works [1][2]. In response to the reviewer, we have clarified that some important issues in the learning-based DAG scheduling domain remain unsolved in these recent works, and our paper solved these issues from a different perspective compared to [1][2]. Accordingly, **in the “Introduction” and “Related Work” section of the revised paper, we have added an additional paragraph to describe the unsolved issues and analyze the methodology difference between [1][2] and our method in detail**. In addition, in the “Results and discussion” section, we have added the comparison results with [2], and experiment results on other larger-scaled and different benchmarks in Table 2-4 of the main text and Table 5-7 of Appendix F, as requested by Reviewer **zot3**. The supplemented results show that our method achieves superior optimization performance in most cases, especially in structurally complex workflows (Tables 5–6 in Appendix F.1 of the revised paper), highlighting its stronger ability to handle with scheduling in complex high-performance computation scenarios. Moreover, our method exhibits lower runtime across settings (Figure 4 and Table 4), providing advantages in scheduling for latency-sensitive applications. **We believe that the additional analyses, clarifications, and new experiments substantiate that our study is not an incremental contribution, but represents a meaningful and breakthrough contribution in the one-shot DAG scheduling domain**.

+ Reviewer **c4UY** only provided brief comments on our paper, mainly questioning the novelty and contribution of our method. In response to this, we have made a detailed explanation to distinguish the methodology and contribution between our method and existing studies in the rebuttal to **c4UY**, similar to that in the response to Reviewer **zot3**. Reviewer **c4UY** also inquired whether our method is capable of handling dynamic workflow scheduling or other scheduling problem variants. We believe that our proposed approach can be extended to latency-sensitive dynamic scheduling scenarios and variants due to its low inference complexity. While such extensions are beyond the current paper’s scope, we plan to explore them in future work.

[1] Jeon et al., *Neural DAG Scheduling via One-Shot Priority Sampling*, ICLR 2023.

[2] Qi et al., *Reinforcement learning for one-shot DAG scheduling with comparability identification and dense reward*, NeurIPS 2025.

---

### Meta-Review · Area_Chair_Pf5c · 2026-01-11

**Summary:**

The paper introduces GAA-PtrNet, a novel one-shot DAG scheduling approach based on pointer networks. The initial reviews were mixed, with 2 reviewers (ppbH, vDpe) being positive and 2 (zot3, c4UY) opting for rejection.
The authors performed an elaborate rebuttal. They addressed several issues such as several ablation studies, the inclusion of some new baselines, and extended or refined descriptions to improve the clarity of their approach and experimental setup. The main remaining concerns are on the significance of the contribution (c4UY) and the novelty of the approach (zot3). In my opinion the authors sufficiently motivate their contribution in a structured way, while reviewer c4UY did not sufficiently substantiate the claim of limited novelty. The degree to which all components are novel is debatable.

**Reviewer Concerns:**

Specify which reviewer concerns you think were addressed by the rebuttal, and which you believe are still outstanding.
The main concerns that have been addressed are the insufficient experimental validation (zot3), inappropriate baseline selection (zot3), unclear descriptions of various aspects of the paper (ppbH, vDpe). The main unresolved issue is the extent to which the work adds sufficient contribution to learning-for-scheduling-domain (c4UY, zot3).

**Reviewer Scores:**

Reviewer ppbH might have raised their score to an 8 based on the rebuttal. Reviewer vDpe would have kept their score an 8. Reviewer zot3’s concerns have been partially addressed, leaving it a borderline case between 4 and 6. Reviewer c4UY would have kept their score at a 2 or 4. However, they offered a very short review which lacked substantiation and provided limited opportunity for the authors to persuade them.

---

### Decision · Program_Chairs · 2026-01-26

Reject